# FACT and Ubp10 collaborate to modulate H2B deubiquitination and nucleosome dynamics

Melesse Nune[1], Michael T Morgan[1], Zaily Connell[2], Laura McCullough[2], Muhammad Jbara[3], Hao Sun[3], Ashraf Brik[3], Tim Formosa[2]*, Cynthia Wolberger[1]*

[1]Program in Molecular Biophysics, Biophysics and Biophysical Chemistry, Johns Hopkins University School of Medicine, Baltimore, United States; [2]Department of Biochemistry, University of Utah School of Medicine, Salt Lake City, United States; [3]Schulich Faculty of Chemistry, Technion-Israel Institute of Technology, Haifa, Israel

**Abstract** Monoubiquitination of histone H2B (H2B-Ub) plays a role in transcription and DNA replication, and is required for normal localization of the histone chaperone, FACT. In yeast, H2B-Ub is deubiquitinated by Ubp8, a subunit of SAGA, and Ubp10. Although they target the same substrate, loss of Ubp8 and Ubp10 cause different phenotypes and alter the transcription of different genes. We show that Ubp10 has poor activity on yeast nucleosomes, but that the addition of FACT stimulates Ubp10 activity on nucleosomes and not on other substrates. Consistent with a role for FACT in deubiquitinating H2B in vivo, a FACT mutant strain shows elevated levels of H2B-Ub. Combination of FACT mutants with deletion of Ubp10, but not Ubp8, confers increased sensitivity to hydroxyurea and activates a cryptic transcription reporter, suggesting that FACT and Ubp10 may coordinate nucleosome assembly during DNA replication and transcription. Our findings reveal unexpected interplay between H2B deubiquitination and nucleosome dynamics.
DOI: https://doi.org/10.7554/eLife.40988.001

*For correspondence:
tim@biochem.utah.edu (TF);
cwolberg@jhmi.edu (CW)

## Introduction

Eukaryotic chromatin is decorated with a wide range of reversible histone post-translational modifications (PTMs) that regulate all processes that require access to DNA, including transcription, DNA replication and DNA repair (*Bannister and Kouzarides, 2011*; *Bowman and Poirier, 2015*). Actively transcribed genes in all eukaryotes are enriched in monoubiquitinated histone H2B, which plays a non-degradative role in promoting transcription (*Fleming et al., 2008*; *Weake and Workman, 2008*) but whose mechanism of action remains poorly understood. Monoubiquitin is conjugated to H2B-K123 in yeast and H2B-K120 in humans, which lie near the C-terminus of histone H2B (*Robzyk et al., 2000*; *Weake and Workman, 2008*; *West and Bonner, 1980*). Monoubiquitination of histone H2B is highly dynamic (*Henry et al., 2003*), and the cycle of ubiquitination and subsequent deubiquitination is an important checkpoint for transcription elongation (*Batta et al., 2011*). H2B-K123 ubiquitination is required for methylation of histone H3K4 (*Chandrasekharan et al., 2010*; *Dover et al., 2002*) and H3K79 (*Ng et al., 2002*), two other marks correlated with actively transcribed genes. However, H2B-Ub also plays a role in promoting transcription that is independent of cross-talk with histone H3 methylation (*Tanny et al., 2007*). In addition to its role in transcription, H2B-Ub plays a role in replication fork progression, nucleosome assembly during DNA replication (*Ng et al., 2002*) and the DNA damage response (*Giannattasio et al., 2005*; *Moyal et al., 2011*; *Uckelmann and Sixma, 2017*). Dysregulation of histone H2B monoubiquitination has been linked to a variety of cancers (*Cole et al., 2015*; *Espinosa, 2008*; *Hahn et al., 2012*).

In the yeast, *Saccharomyces cerevisiae*, histone H2B-K123 is monoubiquitinated by the E2/E3 pair, Rad6/Bre1 (*Hwang et al., 2003*; *Robzyk et al., 2000*; *Wood et al., 2003*), and deubiquitinated by two deubiquitinating enzymes (DUBs): Ubp8 and Ubp10 (*Daniel et al., 2004*; *Gardner et al., 2005*; *Henry et al., 2003*). Both of these DUBs belong to the Ubiquitin Specific Protease (USP) class of cysteine proteases, which contain a characteristic USP catalytic domain (*Komander et al., 2009*). Ubp10 is a monomeric enzyme whereas Ubp8 is part of a four-protein subcomplex within the SAGA complex called the DUB module, which comprises Ubp8, Sgf11, Sus1, and Sgf73 (*Henry et al., 2003*; *Köhler et al., 2010*; *Samara et al., 2010*). Both yeast H2B-Ub DUBs are conserved in humans. USP36, the human homologue of Ubp10, can complement the effects of a *ubp10* deletion on global H2B-Ub in yeast (*Reed et al., 2015*) and USP22, the homologue of Ubp8, is a subunit of human SAGA (*Zhang et al., 2008*). Yeast in which both Ubp10 and Ubp8 have been deleted showed a synergistic increase in the steady-state levels of global H2B-Ub, as well as growth defects (*Emre et al., 2005*). While the roles of Ubp10 and Ubp8 in regulating H2B deubiquitination are well-established, their respective contributions to chromatin-mediated processes are poorly understood.

Despite their shared substrate specificity, Ubp8 and Ubp10 appear to play distinct roles in vivo. Several studies have shown that SAGA/Ubp8 primarily acts on H2B-Ub near promoters and transcription start sites to promote transcription initiation by RNA polymerase II (*Batta et al., 2011*; *Daniel et al., 2004*; *Schulze et al., 2011*). Ubp10 was first identified for its role in regulating subtelomeric gene silencing (*Emre et al., 2005*; *Gardner et al., 2005*; *Kahana and Gottschling, 1999*) and is recruited to silenced chromatin (*Gardner et al., 2005*). However, deletion of *UBP10* alters expression of hundreds of yeast genes as well as H2B ubiquitination genome-wide (*Gardner et al., 2005*; *Orlandi et al., 2004*; *Schulze et al., 2011*), indicating that Ubp10 plays a global role beyond its function in subtelomeric transcriptional repression. Deletion of *UBP8* also alters transcription of several hundred genes (*Gardner et al., 2005*), although an analysis of the data shows little correlation between the genes whose expression is impacted by *ubp8* versus *ubp10* deletion (*Figure 1*). The different impacts on transcription profiles suggest that these two H2B-Ub DUBs have distinct genomic targets. However, SAGA/Ubp8 was recently shown to be involved in transcription of all RNA polymerase II genes (*Baptista et al., 2017*; *Warfield et al., 2017*) and Ubp10 has been found in association with RNA polymerase II (*Mao et al., 2014*), suggesting that both DUBs may at least be present at all genes. A partial resolution of this conundrum comes from a genome-wide ChIP-on-chip study of H2B-Ub in *ubp10* and *ubp8* deletion strains (*Schulze et al., 2011*) which shows that loss of *UBP8* results in an enrichment of H2B-Ub primarily near transcription start sites (TSS), whereas a *ubp10* deletion strain shows broader enrichment of H2B-Ub in mid-coding regions of longer transcription units. The ChIP results suggest that Ubp8 and Ubp10 are required during transcription, but at different times and in different genic locations. However, it remains unclear how each of these factors produces these distinct profiles and what roles each enzyme plays during these processes.

Ubiquitination of histone H2B has been reported to assist recruitment of the histone chaperone, FACT (Facilitates Chromatin Transcription) to active chromatin (*Fleming et al., 2008*). The yeast FACT complex is composed of a heterodimer of Spt16 and Pob3 that is assisted in vitro and in vivo by the DNA binding protein, Nhp6 (*Brewster et al., 1998*; *Ruone et al., 2003*; *Schlesinger and Formosa, 2000*; *Wittmeyer and Formosa, 1995*; *Wittmeyer et al., 1999*). FACT is reported to evict H2A/H2B heterodimers in front of the transcription machinery (*Reinberg and Sims, 2006*) and reassemble the heterodimers in the wake of RNA polymerase II to prevent cryptic transcription initiation (*Fleming et al., 2008*; *Martin et al., 2018*; *Mason and Struhl, 2003*; *Pavri et al., 2006*). The disruption of the H2B ubiquitination cycle or a mutation in the FACT subunit, Spt16, causes a defect in Pol II elongation (*Fleming et al., 2008*). In addition to roles in transcription, FACT and H2B-Ub are each also implicated in DNA replication (*Formosa, 2012*; *Kurat et al., 2017*; *Trujillo and Osley, 2012*). H2B-Ub at replication origins is thought to stabilize the parental nucleosomes after the passage of DNA polymerase (*Trujillo and Osley, 2012*). FACT and H2B-Ub play an important role in the progression of DNA replication, likely by maintaining chromatin stability and orchestrating nucleosome assembly on newly-synthesized DNA (*Lin et al., 2014*; *Trujillo and Osley, 2012*). It is clear that both FACT and H2B-Ub play a pivotal role in stabilizing and assembling nucleosomes in the wake of polymerases during replication and transcription. However, it is not known how FACT and H2B-Ub status affect one another to perform these functions.

We report here a novel role for the histone chaperone, FACT, in stimulating the H2B deubiquitination activity of Ubp10 on nucleosomes. We show that the rate of deubiquitination of yeast H2B-

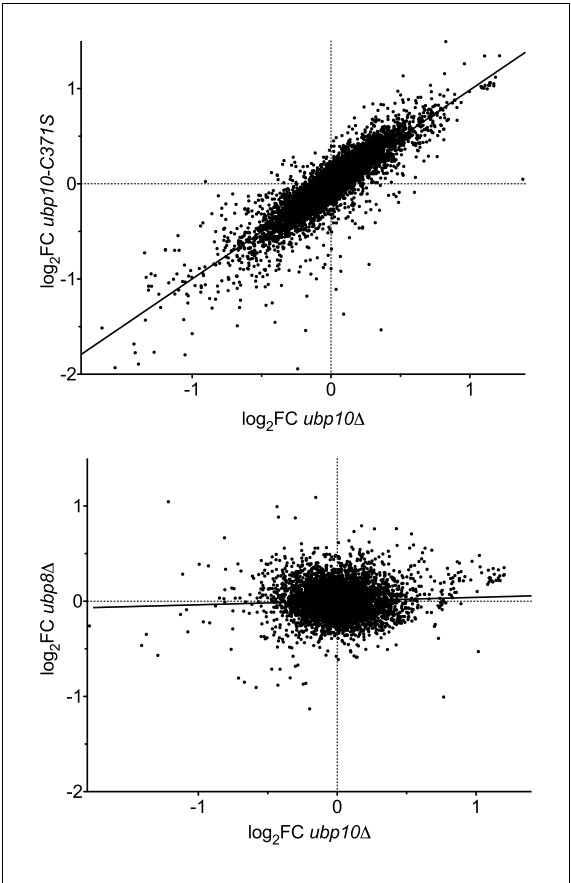

**Figure 1.** Deletion of the *UBP8* and *UBP10* genes have different effects on transcription programs. Analysis of transcription data from *Gardner et al., 2005*. Scatter plots of the $\log_2$ fold change in transcript level relative to WT ($\log_2$FC) are shown for (top panel) a catalytically dead allele (*ubp10-C371S*) vs a deletion (*ubp10Δ*) to demonstrate reproducibility of the array, and (bottom panel) a *ubp8Δ* strain compared with *ubp10Δ*. The two null mutants give a strong correlation (Pearson correlation coefficient r = 0.86, linear regression $R^2$ = 0.74, m = 0.99), validating the reproducibility of the arrays. Deleting *UBP8* affected the transcription of different genes, resulting in poor correlation with *ubp10Δ* (r = 0.055, $R^2$ = 0.0031, m = 0.039).
DOI: https://doi.org/10.7554/eLife.40988.002

Ub is slower when incorporated into nucleosomes as compared to free H2A/H2B-Ub heterodimers, but that the addition of FACT reverses this block. This behavior is in marked contrast to the Ubp8/DUB module, which has robust activity on both heterodimers and intact nucleosomes and is not affected by FACT (*Morgan et al., 2016*). We show that a yeast strain with a FACT deficiency has elevated levels of H2B-Ub, indicating that FACT also stimulates deubiquitination of H2B in vivo. Deleting Ubp10, but not Ubp8, from a strain with mutated FACT conferred strong sensitivity to the DNA replication toxin, hydroxyurea (HU), and activated a cryptic transcription reporter. Our findings suggest that the differential effects of Ubp10 and Ubp8 on the distribution of H2B-Ub result from a global role for Ubp10 and FACT versus a local role of Ubp8/SAGA at promoters and transcription start sites. These observations have important implications for the way in which cycles of H2B ubiquitination and deubiquitination regulate nucleosome dynamics during transcription and DNA replication.

## Results

### Ubp10 preferentially deubiquitinates free yeast H2A/H2B-Ub relative to nucleosomes

During transcription, nucleosomes are at least partially disassembled in order to enable RNA polymerase to access the DNA template and are then reassembled in the wake of the transcribing polymerase. It is not known when during this process ubiquitin is conjugated to histone H2B and when it is removed by either Ubp8/SAGA or Ubp10. Since histone H2A/H2B heterodimers can be ejected and re-inserted during the dynamic nucleosome disassembly and reassembly that accompanies passage of RNA polymerase, it is formally possible that H2B is deubiquitinated when it is in an intact nucleosome, after ejection to the free H2A/H2B-Ub dimer form, or when the nucleosome is in an intermediate state of disassembly or assembly. We previously reported that the Ubp8/SAGA DUB module deubiquitinates H2B in the context of both the nucleosome and the free H2A/H2B-Ub heterodimers, with a modest preference for nucleosomes (*Morgan et al., 2016*). Those results suggested that Ubp8/SAGA could deubiquitinate H2B at any point during transcription.

Since Ubp10 has been reported to associate with RNA polymerase II (*Mao et al., 2014*) and to deubiquitinate H2B in gene bodies (*Schulze et al., 2011*), we asked whether this monomeric DUB discriminates between H2B-ubiquitinated nucleosomes and ubiquitinated H2A/H2B-Ub heterodimers. Using an intein-based semisynthetic approach, we generated ubiquitinated yeast histone H2B in which the C-terminus of ubiquitin was linked to H2B-K123 via a native isopeptide linkage (*Jbara et al., 2018*; *Maity et al., 2016*). This H2B-Ub was used to reconstitute nucleosomes and H2A/H2B-Ub heterodimers. Remarkably, Ubp10 cleaved ubiquitin from H2A/H2B-Ub heterodimers at least 100-fold faster than from nucleosomes containing H2B-Ub (*Figure 2A*). Under the conditions tested, the majority of H2A/H2B-Ub was consumed in less than 5 min while almost all of the NCP-Ub remains uncleaved after 60 min. Similar behavior was recently observed in experiments using human histones containing a cleavable analogue of a native isopeptide linkage and a GST-Ubp10 fusion (*Zukowski et al., 2018*). Taken together, our results indicate that Ubp10 discriminates between freestanding histone heterodimers and those in nucleosomes (*Figure 2A*) whereas Ubp8/SAGA does not (*Morgan et al., 2016*).

### FACT stimulates Ubp10 DUB activity on nucleosomes

In cells, core histones that are not incorporated into nucleosomes are usually bound by histone chaperones, which bind to H2A/H2B heterodimers or H3/H4 heterodimers or heterotetramers (*Elsässer and D'Arcy, 2012*). The histone chaperone, FACT, binds to H2A/H2B heterodimers and facilitates heterodimer eviction and exchange, as well as nucleosome reassembly (*Fleming et al., 2008*; *Mason and Struhl, 2003*; *Orphanides et al., 1998*; *Orphanides et al., 1999*; *Saunders et al., 2003*). In light of the reported functional interaction between H2B-Ub and FACT (*Fleming et al., 2008*; *Pavri et al., 2006*) and the role of FACT in binding both H2A/H2B heterodimers and intact nucleosomes, we asked whether Ubp10 can remove ubiquitin from ubiquitinated yeast H2A/H2B heterodimers or nucleosomes when they are bound to FACT. We unexpectedly found that FACT dramatically increases the rate at which Ubp10 cleaves H2B-Ub in nucleosomes (*Figure 2B–C*). Essentially all of the H2B-Ub in the nucleosomal sample was cleaved in under 60 min in the presence of FACT (*Figure 2B–C*), whereas less than 10% was consumed during the same time period in the absence of FACT (*Figure 2A–C*). By contrast, the addition of FACT had no effect on the rate at which Ubp10 deubiquitinated H2B in H2A/H2B-Ub heterodimers (*Figure 2A–B*). We verified that purified FACT on its own has no DUB activity against the ubiquitinated nucleosome (*Figure 2—figure supplement 1*). To further confirm the dependence of Ubp10 DUB activity on the presence of FACT, we assayed Ubp10 deubiquitination activity on nucleosomes at a fixed time point in the presence of increasing concentrations of FACT. As shown in *Figure 2D*, the amount of nucleosomal H2B-Ub cleaved increases as a function of increasing FACT concentration. Notably, the dose response for FACT in this assay closely matches the affinity of FACT for nucleosomes (*Ruone et al., 2003*; *Winkler et al., 2011*). Interestingly, the ability of FACT to stimulate of H2B deubiquitination does not require Nhp6, as Spt16/Pob3 alone efficiently stimulates the activity of Ubp10 (*Figure 2—figure supplement 2*). These results show that FACT stimulates Ubp10 DUB activity, and that this stimulatory effect is specific to nucleosomal H2B-Ub substrates.

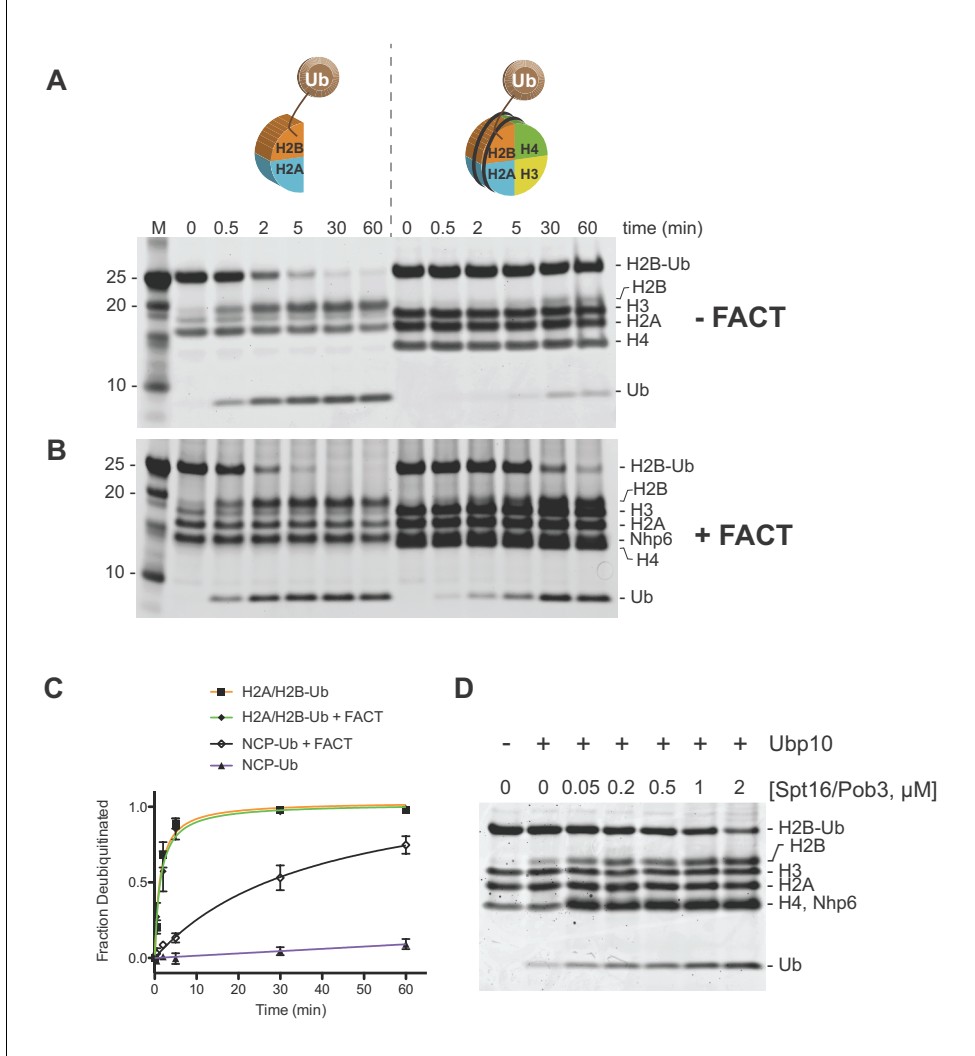

**Figure 2.** Ubp10 preferentially deubiquitinates H2A/H2B-Ub over nucleosomal (NCP) H2B-Ub. (**A–B**) Comparison of Ubp10 activity on H2A/H2B-Ub and NCP-Ub in the absence (**A**) and presence of FACT (**B**). In panel A, 1 µM NCP-Ub and 2 µM H2A/H2B-Ub were incubated with 5 nM Ubp10 and time points were taken by quenching with SDS sample buffer. (**B**) FACT stimulates Ubp10 activity on NCP-Ub. Ubp10 activity was measured as in A, but in the presence of FACT subunits, 2 µM Spt16/Pob3-WT and 2 µM Nhp6. (**C**) The fraction of total substrate consumed over time from assays performed in A-B is shown. The plot was generated by averaging the relative intensity of H2B-Ub bands as compared with uncleaved H2B-Ub at t = 0 from three independent experiments (mean normalized band intensity and standard deviation shown). (**D**) Increasing the concentration of FACT increases the activity of Ubp10. Enzyme activity was monitored by mixing 1 µM NCP-Ub, 2 µM Nhp6, and the indicated concentrations of Spt16/Pob3 in the presence of 5 nM Ubp10. Each reaction was quenched at 60 min.
DOI: https://doi.org/10.7554/eLife.40988.003

The following figure supplements are available for figure 2:

**Figure supplement 1.** FACT alone does not deubiquitinate H2B-Ub.
DOI: https://doi.org/10.7554/eLife.40988.004

**Figure supplement 2.** FACT does not require Nhp6 to stimulate H2B deubiquitination.
DOI: https://doi.org/10.7554/eLife.40988.005

**Figure supplement 3.** Ubp10 activity on ubiquitinated dimers is unaffected by the presence of 601 DNA.
DOI: https://doi.org/10.7554/eLife.40988.006

**Figure supplement 4.** The presence of FACT does not alter nucleosome integrity during H2B deubiquitination.
DOI: https://doi.org/10.7554/eLife.40988.007

A possible explanation for the observed stimulatory effect of FACT is that it alters nucleosomal structure, making it a better substrate for Ubp10. Previous studies have shown that FACT binding can destabilize canonical nucleosomes, disrupting the octamer/DNA contacts, which could result in displacement of H2A/H2B heterodimers (*Belotserkovskaya et al., 2003*; *Chen et al., 2018*; *McCullough et al., 2011*), thereby providing better substrates for Ubp10 (*Figure 2*). During nucleosome reorganization induced by FACT, surfaces of H2A/H2B heterodimers that are buried in the context of the nucleosome become more accessible (*Kemble et al., 2015*) even while the components remained tethered together (*Wang et al., 2018*; *Xin et al., 2009*; *Yang et al., 2018*), which could enhance accessibility of H2B-Ub for deubiquitination by Ubp10 without dimer eviction. We therefore tested the simple model in which binding of DNA to H2A/H2B heterodimers creates a barrier to Ubp10 deubiquitination of H2B-Ub. However, the activity of Ubp10 on H2A/H2B-Ub heterodimers was similar in the presence and absence of 601 DNA (*Figure 2—figure supplement 3*). This experiment does not rule out steric hindrance by DNA in the context of the full histone octamer, but leaves open the possibility that reorganization of the nucleosome by FACT exposes surfaces more favorable to Ubp10 docking on H2B-Ub. Displacement of H2A/H2B-Ub from nucleosomes does not appear to be required, as deubiquitination by Ubp10 was stimulated by FACT without altering the integrity of nucleosomes as judged by native gel electrophoresis (*Figure 2—figure supplement 4*). Moreover, Nhp6 is required for full reorganization of nucleosomes by FACT (*McCullough et al., 2018*) but did not affect the nucleosomal forms produced after deubiquitination.

The finding that FACT stimulates Ubp10 to deubiquitinate nucleosomes stands in stark contrast to the Ubp8/SAGA DUB module. We previously found (*Morgan et al., 2016*) that the abillity of the Ubp8/DUB module to deubiquitinate nucleosomes is not affected by the addition of FACT. SAGA/Ubp8 can therefore access H2B-K123Ub in the context of the nucleosome, but Ubp10 deubiquitinates nucleosomal H2B-Ub poorly without the assistance of FACT.

## FACT stimulation does not correlate with Ubp10 nucleosome-binding activity

Ubp10 contains an unstructured region rich in Asp/Glu that is N-terminal to the catalytic USP domain (residues 362–733) (*Reed et al., 2015*) (*Figure 3A*). The N-terminal unstructured region contains residues that interact with the Sir3/Sir4 silencing proteins and recruit Ubp10 to subtelomeric regions (*Emre et al., 2005*; *Gardner et al., 2005*; *Zukowski et al., 2018*). However, it is not known how Ubp10 is recruited to ubiquitinated nucleosomes elsewhere in the genome. We therefore asked whether Ubp10 alone can bind to nucleosomes in which ubiquitin is linked to H2B-K123 via a non-hydrolyzable linker (*Morgan et al., 2016*). We detected binding of Ubp10 to ubiquitinated nucleosomes in an electrophoretic mobility shift assay (EMSA), with half-maximal binding observed at approximately 0.4 µM Ubp10 (*Figure 3B*). Ubp10 bound with similar apparent affinity to unmodified nucleosomes (*Figure 3—figure supplement 1*), indicating that interactions between the ubiquitin and the Ubp10 catalytic domain do not play a significant role in the observed binding. Deletion of the N-terminal 156 residues had little effect on the affinity of Ubp10 for ubiquitinated nucleosomes (*Figure 3B*). However, a further deletion of the N-terminal 199 residues, Ubp10-(200-792), reduced binding substantially, and no binding was detected with Ubp10 residues 250–792 (*Figure 3B*). The Sir3/Sir4 interaction domain has been mapped to Ubp10 residues 109–133 (*Reed et al., 2015*) so these results show that the Asp/Glu rich region (residues 157–250) of Ubp10 is important for the observed binding to nucleosomes but the Sir3/Sir4 binding site is not.

To determine whether Ubp10 domains that are required for nucleosome binding are also required for DUB activity as well as for stimulation by FACT, we tested the DUB activity of the Ubp10 N-terminal truncations. All three N-terminal truncation mutants were active on H2A/H2B-Ub heterodimers, although the Δ200 and Δ250 truncations had slightly lower activity than intact Ubp10 (*Figure 3C*). Similar to the full length protein, all three truncations displayed weak activity on a nucleosomal substrate in the absence of FACT and enhanced DUB activity on nucleosomes in the presence of FACT (*Figures 3D* and *2A*). Therefore, the truncations that decreased Ubp10 affinity for nucleosomes (Ubp10 200–792 and Ubp10 250–792) were still stimulated by FACT. We also tested the hypothesis that the presence of FACT might enhance binding of Ubp10 to nucleosomes, which could provide a mechanism by which FACT enhances cleavage of nucleosomal H2B-Ub. Gel mobility shift assays for binding to ubiquitinated nucleosomes, however, did not show enhanced affinity of Ubp10 for ubiquitinated nucleosomes in the presence of FACT (*Figure 3—figure supplement 2*).

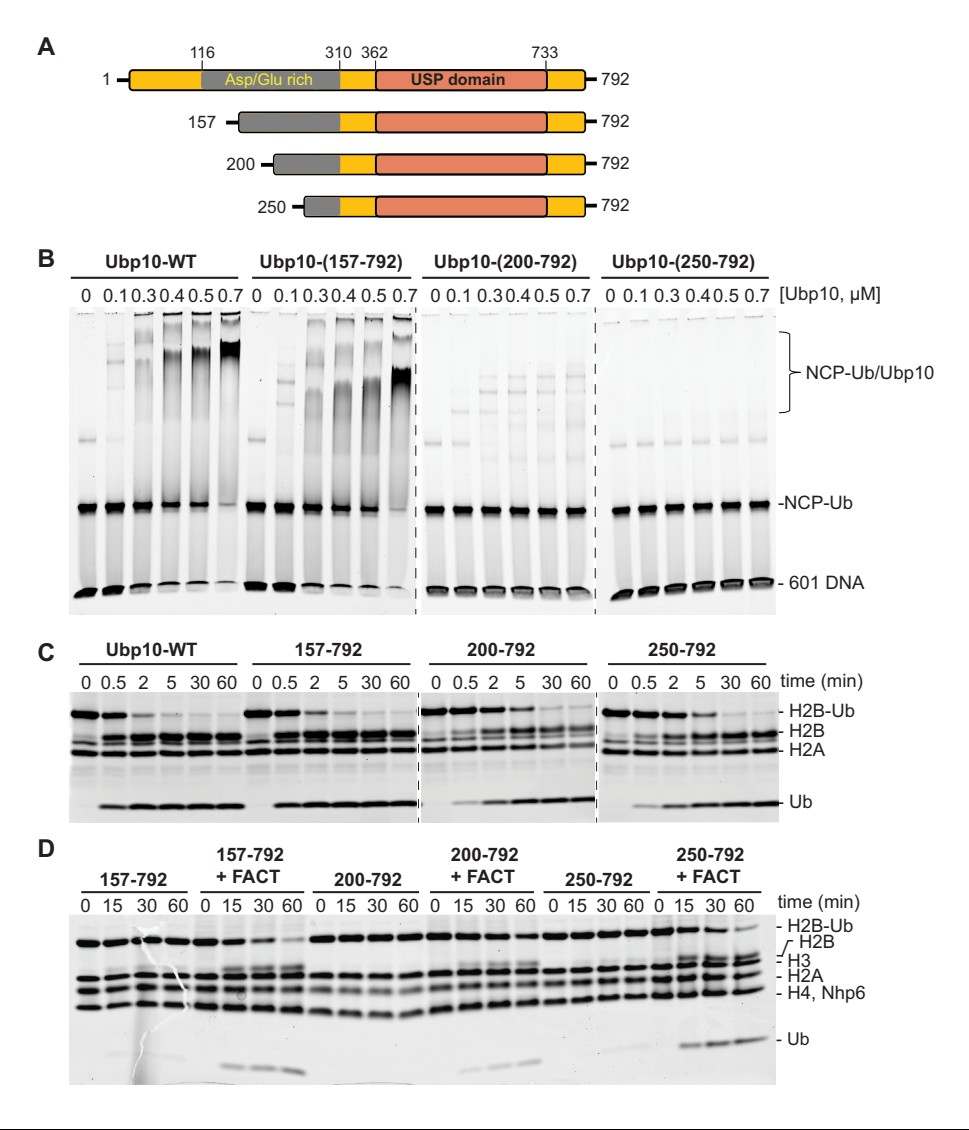

**Figure 3.** FACT stimulates Ubp10 constructs that lack intrinsic ability to bind nucleosomes. (**A**) Schematic of Ubp10 and its truncations with the locations of the catalytic domain (USP) and an N-terminal region rich in aspartic acid and glutamic acid repeats shown. (**B**) Native gel showing complexes formed between Ubp10 constructs and ubiquitinated nucleosomes (purified proteins used in this experiment are shown in *Figure 3—figure supplement 4*). (**C**) Ubp10 activity was measured as in *Figure 2*; time points were taken after incubating 2 μM H2A/H2B-Ub dimers with 5 nM Ubp10 fragments. (**D**) NCP-Ub cleavage activity monitored in the presence and absence of FACT and several Ubp10 constructs.

DOI: https://doi.org/10.7554/eLife.40988.008

The following figure supplements are available for figure 3:

**Figure supplement 1.** Native gel showing binding of several Ubp10 fragments to unmodified yeast nucleosomes.
DOI: https://doi.org/10.7554/eLife.40988.009

**Figure supplement 2.** FACT does not affect the affinity of Ubp10 for nucleosomes.
DOI: https://doi.org/10.7554/eLife.40988.010

**Figure supplement 3.** Binding of the SAGA DUB module to ubiquitinated nucleosomes is not detectable by EMSA.
DOI: https://doi.org/10.7554/eLife.40988.011

**Figure supplement 4.** SDS-PAGE gel showing purity of Ubp10 truncations.
DOI: https://doi.org/10.7554/eLife.40988.012

These results suggest that the ability of FACT to stimulate Ubp10 is not coupled to the intrinsic ability of Ubp10 to bind nucleosomes. We speculate that Ubp10 binds to nucleosomes primarily through the histone dimers but remains poised for deubiquitination until FACT acts on the nucleosomes.

## A FACT mutant strain has elevated levels of H2B-Ub

Our in vitro assays show that the ability of Ubp10 to deubiquitinate nucleosomes is greatly enhanced in the presence of the histone chaperone FACT (*Figure 2*). If FACT activity also stimulates Ubp10 activity in vivo, a defect in FACT activity should phenocopy the effects of a Ubp10 deletion. To test this, we compared the relative ratio of monoubiquitinated H2B to unmodified H2B in a yeast strain with the *pob3-L78R* mutation, which destabilizes the Pob3 subunit of FACT and reduces its level by about 10-fold under permissive growth conditions (*Schlesinger and Formosa, 2000*; *VanDemark et al., 2008*). This causes defects in both transcription and DNA replication (*Schlesinger and Formosa, 2000*). As shown in *Figure 4*, the *pob3-L78R* strain had an elevated level of H2B-Ub (1.9-fold increased) that is comparable to that in a *ubp10* deletion strain (1.4-fold increased) when normalized for total H2B. This increase in H2B-Ub in a FACT mutant is consistent with a role for FACT in deubiquinating H2B-Ub in vivo. A strain lacking Ubp10 and also carrying the FACT defect had roughly the same increased ratio of H2B-Ub as the *pob3-L78R* mutant (2.0-fold increased, *Figure 4B*), consistent with the interpretation that FACT and Ubp10 cooperate to deubiquitinate H2B in the same pathway. Together, these results show that FACT activity can contribute to H2B deubiquitination in vivo, supporting the physiological relevance of the in vitro data showing that FACT is stimulates Ubp10 activity on nucleosomes (*Figure 2D*).

Our results differ from a previous study (*Fleming et al., 2008*) that found that cells with a temperature-sensitive *SPT16* allele (*spt16-197*) showed a decrease in FLAG-H2B-Ub levels when shifted to a restrictive temperature. In that experiment, cells initially experienced normal levels of FACT, followed by an acute reduction of the essential FACT complex to a level that does not support viability (*Malone et al., 1991*). By contrast, the *pob3-L78R* mutant used in this study experiences chronically low levels of FACT but is viable under the conditions tested (*Schlesinger and Formosa, 2000*). The use of an allele that destabilizes a different subunit of FACT under conditions of chronic rather than acute exposure to FACT depletion may explain the different outcomes, and may suggest clues regarding the mechanism through which FACT affects Ubp10 activity in vivo.

## FACT and Ubp10 cooperate during DNA replication

In addition to their roles in transcription (*Batta et al., 2011*; *Fleming et al., 2008*; *Pavri et al., 2006*; *Reinberg and Sims, 2006*; *Weake and Workman, 2008*), both FACT and H2B monoubiquitination have been implicated in assembling and stabilizing nucleosomes during DNA replication (*Jasencakova and Groth, 2010*; *Lin et al., 2014*; *Trujillo and Osley, 2012*). FACT has been proposed to play an important role in DNA replication by assisting in initiation, DNA unwinding, histone eviction, and chromatin reassembly (*Lin et al., 2014*; *Ransom et al., 2010*; *Schlesinger and Formosa, 2000*; *Trujillo and Osley, 2012*). H2B monoubiquitination near origins of replication supports replisome stability, fork progression and checkpoint pathways (*Lin et al., 2014*; *Trujillo and Osley, 2012*). Insights into the role of H2B-Ub in DNA replication come from studying the effects of deleting the H2B E3 ligase, Bre1 (*Lin et al., 2014*; *Trujillo and Osley, 2012*), or mutants expressing H2B with a K123R substitution, which cannot be ubiquitinated (*Lin et al., 2014*; *Trujillo and Osley, 2012*). The role of H2B deubiquitinating enzymes in DNA replication, however, has not been explored.

In light of our finding that FACT stimulates Ubp10 activity on nucleosomes in vitro and is required to maintain wild type levels of H2B ubiquitination in vivo, we asked whether FACT functionally interacts with Ubp10 during DNA replication. We tested the sensitivity of Ubp10 deletions and FACT mutants to hydroxyurea, which depletes cellular dNTPs and induces replication fork stalling (*Singh and Xu, 2016*). The *pob3-L78R* FACT mutant has previously been shown to display a severe growth defect at 30° (temperature sensitivity) but is viable at 25°C (*Schlesinger and Formosa, 2000*), while the *ubp10Δ* strain has a mild growth defect and is not temperature sensitive (*Gardner et al., 2005*). Growth of the *ubp10Δ/pob3-L78R* double mutant at 25°C was only mildly reduced relative to the single mutants (*Figure 5A*). Neither the *ubp10Δ* nor the *pob3-L78R* FACT mutant was sensitive to 90 mM hydroxyurea (*Figure 5A*). However, the double *ubp10Δ/pob3-L78R*

mutant displayed a significant growth defect under these or milder conditions, with a synthetic defect observed even at levels of HU as low as 6 mM (*Figure 5A*). To test whether this synthetic growth defect in the presence of HU is unique to the *pob3-L78R* FACT mutant, we also examined the genetic interaction between a *ubp10* deletion and two other FACT mutants, *spt16-11* and *pob3-Q308K*. As shown in *Figure 5B*, both the *spt16-11* and *pob3-Q308K* mutants displayed synthetic growth defects in the presence of HU when combined with a *ubp10* deletion.

Ubp8/SAGA, in contrast to Ubp10, does not depend upon FACT to deubiquitinate nucleosomes in vitro (*Morgan et al., 2016*). We therefore predicted that *ubp8* and FACT should not have the same genetic interaction as *ubp10* and FACT. Indeed, a *ubp8Δ/pob3-L78R* double mutant did not show any synthetic growth defects in the presence of hydroxyurea, even at concentrations as high as

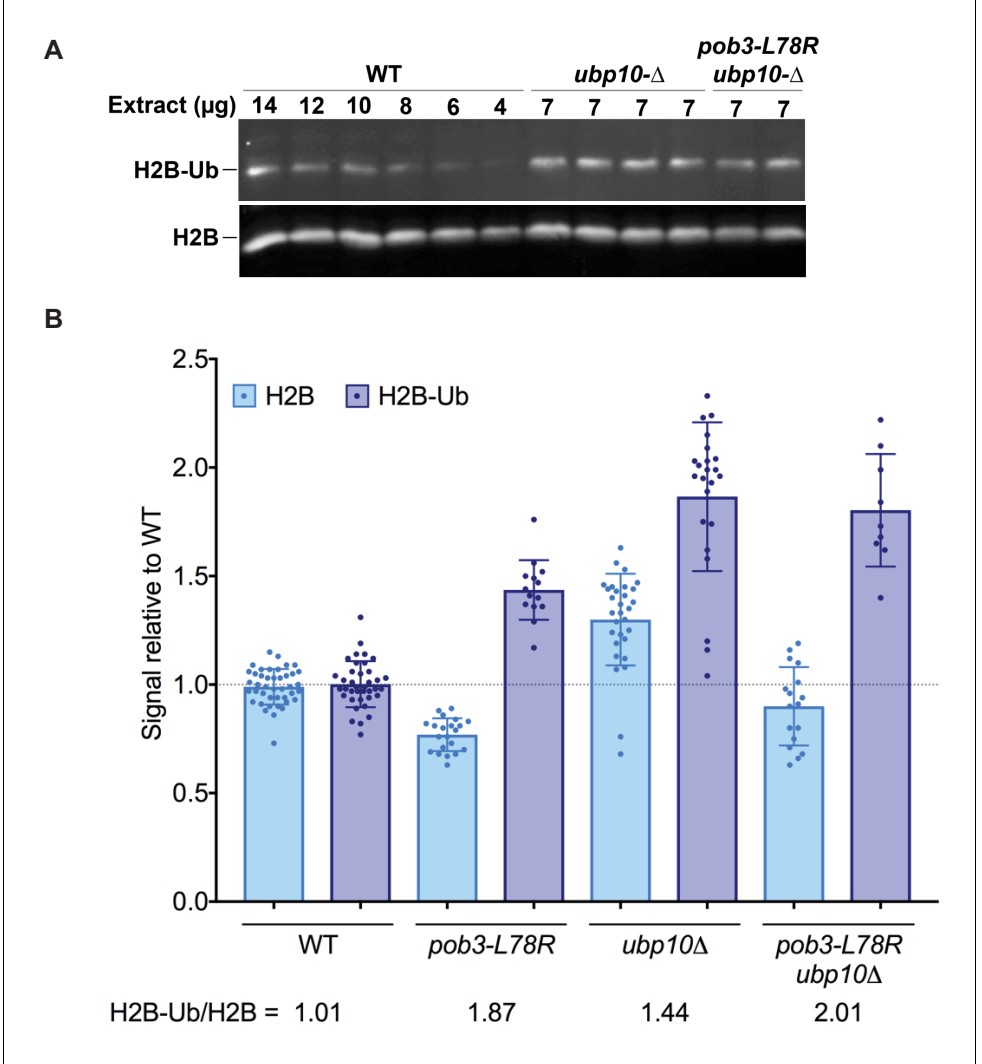

**Figure 4.** H2B-Ub levels are elevated in a FACT mutant strain. (**A**) Representative western blot analysis of TCA extracts from *WT*, *ubp10Δ*, and *ubp10Δ pob3-L78R* strains probed with antibodies against H2B-Ub and re-probed with antibodies against H2B. (**B**) Relative steady-state levels of H2B-Ub for *WT*, *pob3-L78R*, *ubp10Δ*, and *ubp10Δ/ pob3-L78R* strains (*Table 1*). The average and standard deviation from multiple biological replicates is shown. The numbers at the bottom indicate the relative H2B-Ub increase normalized to the paired unmodified H2B value (H2B-Ub/H2B) within each individual gel. Total H2B is decreased in the mutant because the slow growth of the strain leads to a larger average cell size and therefore a lower contribution of nuclear proteins to the total protein level that was used to normalize loading.

DOI: https://doi.org/10.7554/eLife.40988.013

150 mM (*Figure 5C*). These observations point to a specific cooperative function of Ubp10 and FACT in DNA replication that cannot be performed by Ubp8.

## FACT and Ubp10 cooperate to suppress cryptic transcription

Both FACT and H2B ubiquitination are needed to maintain wild type levels of nucleosome occupancy (*Feng et al., 2016*; *Fleming et al., 2008*; *Jamai et al., 2009*). Defects in nucleosome occupancy can give rise to altered transcription patterns and activation of cryptic transcription initiation in gene coding regions (*Fleming et al., 2008*; *Kaplan et al., 2003*). Mutations in FACT cause expression of the *lys2-128∂* Spt⁻ phenotype reporter (Suppression of Ty1 insertion) (*Brewster et al., 1998*; *Malone et al., 1991*; *Schlesinger and Formosa, 2000*; *Simchen et al., 1984*), which reveals failure to establish normal chromatin-mediated repression of this promoter (*Cheung et al., 2008*; *Kaplan et al., 2003*). Wild type strains with this reporter have normal chromatin and do not grow on medium lacking lysine, but all three FACT mutants tested here express the reporter and grow (called the Spt⁻ phenotype; *Figure 6*). This readout of transcription initiation resulting from poor quality chromatin was not affected by the loss of Ubp10 (*Figure 6A,B*; the growth defects caused by combining *ubp10Δ* with FACT mutations were similar on -lys and complete media so they are not due to changes in the Spt⁻ phenotype) or loss of Ubp8 (*Figure 6A* and not shown). We then tested the effect of *ubp10Δ* on expression of a cryptic promoter reporter, which supports growth in the absence of histidine if galactose-induced transcription of the reporter activates a cryptic promoter within the *FLO8* gene (*Cheung et al., 2008*) (*Figure 6D*). While neither a *ubp10Δ* nor an *spt16-11* mutant activated the reporter gene, the double *ubp10Δ/spt16-11* mutant displayed significant levels of cryptic promoter activation (compare Gal Complete and Gal -his in *Figure 6C*) whereas the double *ubp8Δ/spt16-11* mutant did not. We were not able to see an effect of combining a *ubp10Δ* with

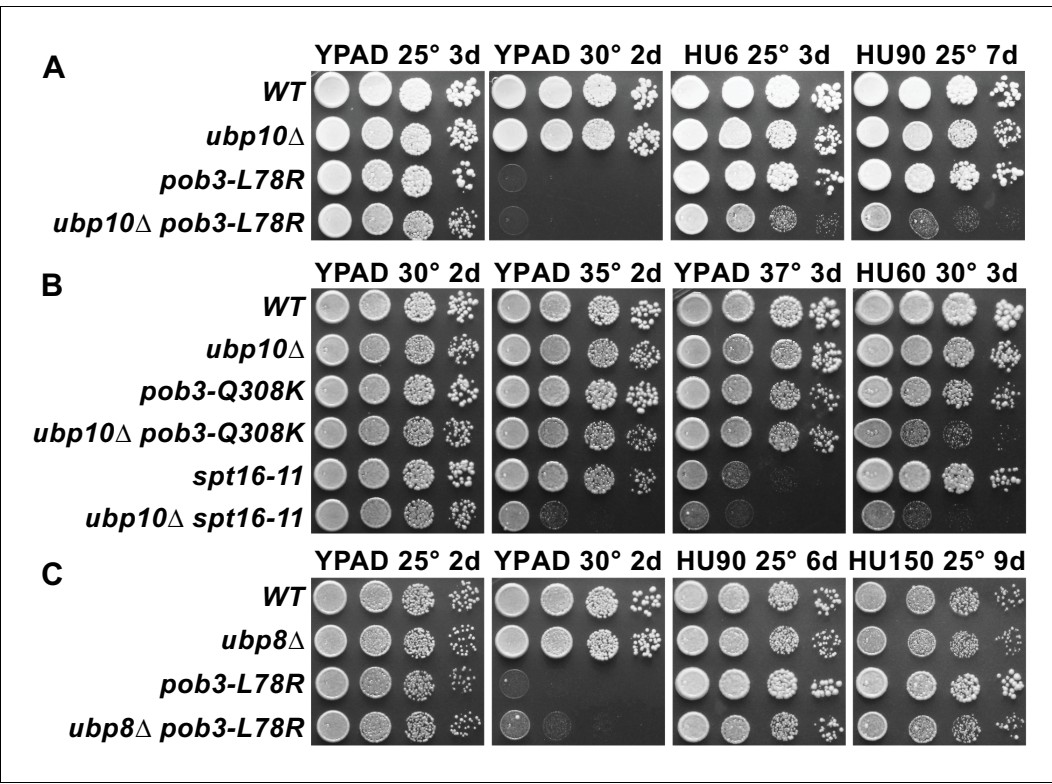

**Figure 5.** Combining *ubp10Δ* and FACT alleles causes sensitivity to HU. (A–C) Strains indicated (*Table 1*) were grown to saturation, then 10-fold serial dilutions were spotted on rich medium (YPAD) with and without the indicated concentrations of hydroxyurea (HU, mM). Plates were incubated at the temperature indicated for the time shown (days). Combining *pob3-L78R* with *ubp10Δ* caused HU sensitivity (A), but combining it with *ubp8Δ* did not (C). Combining *ubp10Δ* with other alleles of FACT also caused synthetic defects on HU (B).
DOI: https://doi.org/10.7554/eLife.40988.014

the *pob3-L78R* allele because the *pob3-L78R* mutation on its own robustly activates the cryptic transcription reporter and enables near wild-type levels of growth (*Figure 6C*), again highlighting the different effects of different FACT alleles. Taken together, these results indicate that Ubp10, but not Ubp8, acts in combination with FACT to maintain normal chromatin organization in the wake of RNA polymerase II transcription.

## Discussion

We have discovered a previously uncharacterized synergy between the H2B deubiquitinating enzyme, Ubp10, and the histone chaperone, FACT, that suggests a unifying model for the role of H2B ubiquitination and deubiquitination in nucleosome dynamics. The role of H2B monoubiquitination in recruiting FACT to nucleosomes and the importance of both FACT and H2B-Ub in transcription (*Batta et al., 2011*; *Fleming et al., 2008*; *Weake and Workman, 2008*) and in DNA replication (*Lin et al., 2014*; *Trujillo and Osley, 2012*), are well-established. However, the interplay between nucleosome dynamics, H2B deubiquitination and histone chaperones had not been explored. In this study, we report a role for the histone chaperone, FACT, in deubiquitinating H2B-Ub and show that Ubp10 depends on FACT to efficiently cleave H2B-Ub from nucleosomes (*Figure 2*). This behavior is unlike Ubp8/SAGA, which does not require FACT to deubiquitinate nucleosomes (*Morgan et al., 2016*). Consistent with our in vitro results, we find that a mutation that decreases the level of FACT (*pob3-L78R*) results in increased levels of H2B-Ub that are comparable to those caused by loss of Ubp10 (*Figure 4*). We also show that combining FACT mutations with loss of Ubp10, but not loss of Ubp8, causes sensitivity to hydroxyurea, suggesting a role for the FACT-Ubp10 collaboration in DNA replication. Similarly, combining FACT mutations with deletion of *UBP10*, but not *UBP8*, activates a cryptic promoter reporter gene, reflecting a role for Ubp10 in maintaining nucleosome occupancy during transcription. These combined in vivo and in vitro observations point to a role for Ubp10 and FACT in jointly maintaining chromatin organization and have important implications for the different cellular roles of the two H2B-Ub DUBs, Ubp8 and Ubp10.

Our results are consistent with a model in which Ubp10 plays a global role in regulating nucleosome dynamics in concert with FACT, while Ubp8 plays a more restricted role at sites of transcription initiation. This view is consistent with previous observations that Ubp8 deletion leads to higher levels of H2B-Ub in the vicinity of the +1 nucleosome, whereas Ubp10 deletions exhibit broader enrichment of H2B-Ub in gene bodies, particularly in longer open reading frames (*Schulze et al., 2011*). Ubp8 is targeted to genes in the context of the SAGA complex, a global transcription factor that associates with promoter regions of virtually all RNA polymerase II genes (*Baptista et al., 2017*; *Baptista et al., 2018*; *Bonnet et al., 2014*; *Venters et al., 2011*), which accounts for the observed pattern of H2B-Ub enrichment in *ubp8* deletion strains (*Schulze et al., 2011*). The genome-wide pattern of H2B-Ub enrichment found in *ubp10* deletion strains (*Schulze et al., 2011*) can be explained by a global function for Ubp10 in nucleosome assembly through its partnership with FACT. FACT has been implicated in transcription through its increased association with frequently transcribed genic regions (*Feng et al., 2016*; *Mayer et al., 2010*; *Pathak et al., 2018*; *Saunders et al., 2003*), but these same studies show that it is also abundant in intergenic regions. FACT physically interacts with both the MCM replicative helicase and with DNA polymerase α, and it promotes rapid deposition of nucleosomes in an in vitro replication system (*Wittmeyer and Formosa, 1997*; *Yang et al., 2016*; *Yeeles et al., 2017*). FACT is found in yeast cells at about two-thirds the abundance of nucleosomes and therefore is a global component of chromatin that participates in a broad range of chromatin-dependent processes (*Gurova et al., 2018*). These roles include restoring chromatin integrity after disruptions like transcription (*Martin et al., 2018*) but emerging evidence shows that FACT may have a less prominent role in proliferation of differentiated mammalian cells (*Gurova et al., 2018*; *Shen et al., 2018*). Collectively, these findings suggest, instead, that the primary role of FACT is to stabilize or maintain existing global chromatin architectures and promote transitions to new patterns during differentiation. The finding that Ubp10, but not Ubp8, plays an unanticipated role in DNA replication (*Figures 4* and *5*) and suppression of cryptic transcription (*Figure 6*) that overlaps with the role of FACT supports the idea that, like FACT, Ubp10 has a global function in maintaining stable chromatin architecture against a range of potential perturbations.

H2B ubiquitination has been proposed to stabilize chromatin as judged by a genome-wide reduction in nucleosome occupancy seen in mutants that lack Rad6, the E2 that ubiquitinates H2B, or in

which wild type H2B is replaced by H2B-K123A, which cannot be ubiquitinated (*Batta et al., 2011*). Deletion of *UBP8* enhances nucleosome occupancy genome-wide. However, the simple model that high global levels of H2B-Ub correlate with higher nucleosome occupancy is contradicted by the observation that highly transcribed genes have normal levels of nucleosome occupancy in a *ubp8Δ* mutant, even though these genes have markedly lower nucleosome occupancy in *rad6Δ* and H2B-K123A mutants (*Batta et al., 2011*). Our finding that a double *ubp10Δ/spt16-11* mutant activates a cryptic promoter points to a unique role for Ubp10 acting in concert with FACT to maintain nucleosome organization and is consistent with prior studies that suggest that cycles of both ubiquitination and deubiquitination are critical for nucleosome assembly during transcription (*Batta et al., 2011*; *Fleming et al., 2008*; *Henry et al., 2003*; *Pavri et al., 2006*).

In addition to its role in regulating global levels of H2B-Ub, Ubp10 regulates spreading of heterochromatic silencing at telomeres and at mating type loci (*Emre et al., 2005*; *Gardner et al., 2005*). This function is mediated by the N-terminus of Ubp10, which is recruited by the SIR complex to subtelomeric regions (*Emre et al., 2005*; *Gardner et al., 2005*; *Zukowski et al., 2018*). The silencing and global H2B ubiquitination functions of Ubp10 appear to be separable, as expression of N-terminal truncations of Ubp10 can restore wild type levels of H2B ubiquitination without restoring subtelomeric silencing (*Reed et al., 2015*). A recent study (*Zukowski et al., 2018*) reported that Sir2/Sir4, a

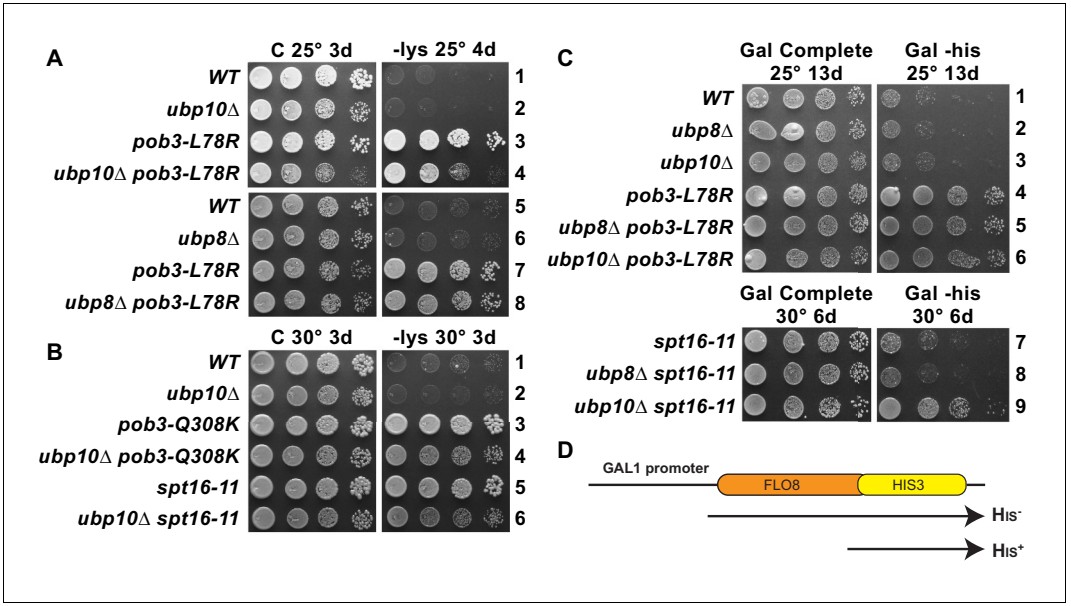

**Figure 6.** Combined effect of FACT mutants and *ubp10Δ* or *ubp8Δ* deletions on the Spt⁻ phenotype and cryptic promoter activation. (A,B) *ubp10Δ* did not affect the Spt⁻ phenotype of FACT mutants. Dilutions of the same strains shown in *Figure 5* were plated on synthetic medium with (C, complete) or without lysine (-lys). These strains with the *lys2-128∂* allele are auxotrophic for lysine, but defects in chromatin integrity allow expression of the gene, which is revealed as growth on -lys (the Spt⁻ phenotype; see *Simchen et al., 1984*). FACT mutants displayed this phenotype, but this was not affected by *ubp8Δ* or *ubp10Δ* (A, B, and not shown). (C) Activation of a cryptic transcription reporter in a *ubp10Δ/spt16-11* mutant strain reveals a defect in restoring chromatin in the wake of RNA Pol II passage. Strains with an out-of-frame fusion of *HIS3* to a site downstream of a cryptic promoter in the *FLO8* gene (panel D, adapted from *Cheung et al., 2008*) are auxotrophic for histidine when the *GAL1* promoter driving transcription of this reporter is repressed on glucose (not shown) but can grow without histidine on synthetic medium containing galactose (Gal -his). Strains with the *pob3-L78R* mutation have this phenotype, indicating activation of the cryptic promoter, masking any potential effects of *ubp8Δ* or *ubp10Δ*. The *spt16-11* allele alone did not activate this reporter but did when combined with *ubp10Δ*.

DOI: https://doi.org/10.7554/eLife.40988.015

The following figure supplement is available for figure 6:

**Figure supplement 1.** Removing the SIR-interaction domain (residues 109–133) from Ubp10 does not affect the phenotypes of a FACT mutant.

DOI: https://doi.org/10.7554/eLife.40988.016

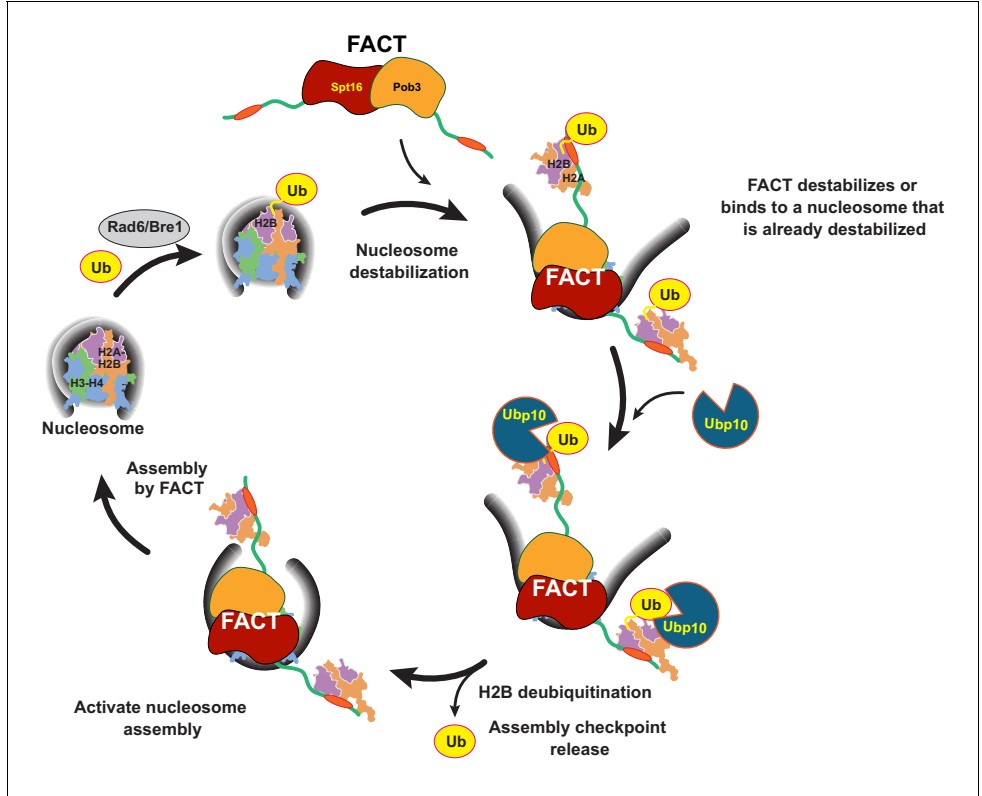

**Figure 7.** Model for coordinated H2B deubiquitination and nucleosome assembly by FACT/Ubp10 during transcription and DNA replication. Rad6/Bre1 ubiquitinates nucleosomal histone H2B during transcription and DNA replication. The presence of H2B-Ub recruits FACT to the nucleosome. FACT destabilizes nucleosomes or binds to nucleosomes that are already destabilized by transcription or replication factors. Ubp10 deubiquitinates H2B-Ub in the context of fully or partially exposed H2A/H2B-Ub heterodimers while still tethered to Spt16/Pob3. Deubiquitination of H2B signals passage of polymerases and deposition of histones in the wake of polymerases. The deubiquitinated nucleosome is reassembled by FACT, followed by dissociation of FACT. The full reorganization depicted here supports Ubp10 activity, but since Spt16-Pob3 heterodimers can support Ubp10 as well, activation of Ubp10 may require only the early stages of reorganization that are not dependent on Nhp6.
DOI: https://doi.org/10.7554/eLife.40988.017

subset of the SIR complex, stimulates the activity of GST-Ubp10 on both human nucleosomes and H2A/H2B-Ub heterodimers containing a hydrolyzable non-native linkage between ubiquitin and H2B. The observed stimulation was attributed to the ability of Sir2/Sir4 to recruit Ubp10 to chromatin, as has also been observed in vivo (*Emre et al., 2005*), as well as to a proposed allosteric stimulation of Ubp10 on both H2A/H2B-Ub heterodimers and on ubiquitinated nucleosomes (*Zukowski et al., 2018*). The effect observed in that study depended on residues 109–133 of Ubp10, which do not play a role in either Ubp10 catalytic activity on yeast H2A/H2B-Ub heterodimers, the ability to be stimulated by FACT (*Figure 3*), or FACT function in vivo (*Figure 6—figure supplement 1*). The effect of Sir2/Sir4 on Ubp10 thus appears to be restricted to its silencing-specific functions. Notably, human Usp15 also preferentially deubiquitinates H2A/H2B-Ub dimers and is stimulated by the splicing factor SART3 (*Long et al., 2014*), although in this case SART3 has the inverse effect: it enhances the activity of Usp15 on H2A/H2B-Ub but not on nucleosomes.

It is not clear why nucleosomes are such a poor substrate for Ubp10 as compared to H2A/H2B-Ub heterodimers. Ubp8 is targeted to nucleosomes in the context of the SAGA DUB module, in which Ubp8 forms a complex with Sgf11, Sus1 and Sgf73 (*Henry et al., 2003*; *Köhler et al., 2010*). The specificity of Ubp8 for H2B-K123 is conferred by Sgf11, which has a zinc finger domain with an arginine-rich patch (*Köhler et al., 2010*; *Samara et al., 2010*) that docks in the nucleosome acidic patch between histones H2A and H2B (*Morgan et al., 2016*). By contrast, Ubp10 is a monomeric enzyme that lacks accessory proteins that could promote binding to nucleosomes. However, Ubp10 has surprisingly high affinity for nucleosomes, binding with an apparent Kd of ~400 nM (*Figure 3B*).

The wild-type Ubp8/DUB module, by contrast, does not bind detectably to nucleosomes by EMSA (*Figure 3—figure supplement 3*). The ability of Ubp10 to bind nucleosomes and the stimulation of Ubp10 enzymatic activity by FACT do not seem to be connected, as N-terminal deletions of Ubp10 that impair nucleosome binding are still stimulated by FACT (*Figure 3B–D*). Moreover, deletions within the N-terminal unstructured region of Ubp10 (*Reed et al., 2015*), which include residues important for recruitment to telomeres by the SIR complex (*Kahana and Gottschling, 1999*), have little or no effect on global H2B ubiquitination (*Gardner et al., 2005*). The significance of the ability of Ubp10 to bind nucleosomes, which was also observed by *Zukowski et al. (2018)*, and the use of nucleosomes as substrates for deubiquitination therefore remains unresolved.

How does FACT stimulate the activity of Ubp10 on nucleosomes? While there is no direct evidence that FACT and Ubp10 work together in vivo, the synthetic genetic interaction and the biochemical cooperation that we observe is consistent with such a possibility. We propose that the structural changes that occur in canonical nucleosomes upon FACT binding are likely key to addressing this question, as the observed stimulation is specific to nucleosomal substrates. FACT can reorganize the nucleosome by disrupting histone/DNA contacts, leaving the nucleosome in an 'open' or 'destabilized' state (*Chen et al., 2018*; *Kemble et al., 2015*; *McCullough et al., 2011*) and by fully displacing H2A/H2B heterodimers (*Belotserkovskaya et al., 2003*; *Chen et al., 2018*; *Hsieh et al., 2013*; *Orphanides et al., 1999*; *Wang et al., 2018*; *Xin et al., 2009*). The most straightforward explanation for the effect of FACT on Ubp10 activity is therefore that FACT increases H2B deubiquitination by evicting H2A/H2B-Ub heterodimers, which are an excellent substrate for Ubp10 (*Figures 2A* and *3C*). However, H2A-H2B displacement has been estimated to be limited to 20–50% of the total heterodimers (*Orphanides et al., 1998*; *Xin et al., 2009*) and requires the addition of Nhp6 to yeast Spt16-Pob3, whereas we observe complete deubiquitination, and no requirement for Nhp6. Dimer displacement can therefore only partially explain the effect of FACT. While it is not known whether ejected H2A/H2B heterodimers retain the H2B-K123 ubiquitin modification in vivo, our in vitro results clearly show that an H2A/H2B-Ub dimer bound to FACT can be rapidly deubiquitinated by Ubp10. Alternatively, FACT may stimulate Ubp10 by partly unwinding the DNA (*Kemble et al., 2015*; *McCullough et al., 2011*; *Valieva et al., 2016*), generating a hexasome, or yielding some other intermediate in nucleosome disassembly that either exposes surfaces that allow Ubp10 to interact more favorably with the H2B-Ub linkage or relieves steric clashes. In this second scenario, Ubp10 deubiquitinates H2B only after FACT has begun to destabilize the nucleosome, but without complete eviction of the dimer. In support of this idea, we find that the nucleosomes are intact following deubiquitination by Ubp10 (*Figure 2—figure supplement 4*). In either scenario, Ubp10 must deubiquitinate H2B-Ub at some point between the time the nucleosome is destabilized and before it is reassembled in the wake of either DNA or RNA polymerase. We think it is unlikely that FACT stimulates Ubp10 by recruiting it to the nucleosome, as we see no evidence that FACT enhances Ubp10 binding (*Figure 3B* and *Figure 3—figure supplement 2*). The enhancement of Ubp10 activity provides the first example of a physiologically relevant substrate that appears to be activated by FACT-mediated destabilization of nucleosomes. Further studies will be needed to unravel the molecular determinants governing Ubp10 substrate preference as well as the mechanism by which FACT activates deubiquitination of nucleosomes.

The coupling of Ubp10 and FACT activity provides a missing link between cycles of H2B ubiquitination and deubiquitination, and FACT activity and nucleosome disassembly and assembly that allows us to propose a global model for the role of H2B ubiquitination in chromatin dynamics (*Figure 7*). Ubiquitination of H2B by Rad6/Bre1 recruits FACT, which can either facilitate nucleosome disassembly or bind to nucleosomes that are already destabilized. Ubp10 then deubiquitinates H2B, enhancing the ability of FACT to promote reassembly of the nucleosome and/or reinsertion of a deubiquitinated H2A/H2B dimer. Deubiquitination of H2B could potentially favor nucleosome reassembly by enhancing release of FACT from its proposed checkpoint function (*McCullough et al., 2018*). The sequence of H2B ubiquitination, FACT-mediated nucleosome reorganization, then deubiquitination by Ubp10 and accelerated assembly could propel sequential assembly of nucleosomes, either in the wake of RNA polymerase or during DNA replication. This proposed sequence of events could also occur in humans, as well, given the conservation of all components of this sytem: RAD6B/RNF20/40 are the human E2/E3, USP36 is the homologue of Ubp10 and human FACT, Spt16/SSRP1, is the homologue of the yeast FACT complex, Spt16/Pob3/Nhp6. Our study provides a framework for understanding how H2B-Ub deubiquitination is coupled to the activity of the histone chaperone,

FACT, in producing dynamic changes to nucleosome structure, and has exciting implications for understanding the mechanism by which dynamic cycles of ubiquitination and deubiquitination regulate chromatin organization.

## Materials and methods

### Protein expression and purification

#### Purification of Ubp10-WT and Ubp10 truncations

To make the full-length wild-type Ubp10 expression plasmid (pMN2), the protein coding sequences were amplified from *Saccharomyces cerevisiae* genomic DNA by PCR using KOD polymerase (EMD Millipore). The amplified product containing an N-terminal His6-tag and TEV (tobacco etch virus) cleavage site was inserted into a vector that contains thioredoxin protein, pET32a, using IN-fusion cloning kit (Clontech). Ubp10 N-terminal deletions containing residues 157–792 (pMN3), 200–792 (pMN4), and 250–792 (pMN5), were similarly amplified from the original Ubp10-WT expression plasmid, pMN2, and inserted into pET32a. Ubp10-containing plasmids were expressed in Rosetta (DE3) cells. Briefly, a starter culture was grown to an OD of 0.6, then transferred to 1 L M9ZB medium and allowed to grow at 37°C. When the OD reached ~1.5–2, the medium was supplemented with 1 mM isopropyl-β-D-thiogalactoside (ITPG) and the temperature was shifted to 20°C for an overnight induction. Pelleted cells were lysed in lysis buffer, 25 mM HEPES pH 7.5, 20 mM imidazole pH 7.5, 600 mM NaCl, 10 mM 2-mercaptoethanol, and 1 mM phenylmethylsulfonyl fluoride (PMSF). The lysate was recovered by centrifugation and the supernatant was loaded onto 5 ml HisTrap HP column (GE Healthcare) using buffer A (25 mM HEPES pH 7.5, 20 mM imidazole pH 7.5, 600 mM NaCl, 10 mM 2-mercaptoethanol). Bound protein was eluted with buffer B (25 mM HEPES pH 7.5, 300 mM imidazole pH 7.5, 600 mM NaCl, 10 mM 2-mercaptoethanol). To cleave the purification tags, 1 mg of TEV protease (per mg of protein) was added to the combined fractions and dialyzed overnight against buffer A. The dialyzed sample was then reloaded onto a HisTrap column to remove the cleaved purification tag. The protein was then diluted with ion exchange binding buffer (25 mM HEPES pH 7.5, 50 mM NaCl, 10 mM 2-mercaptoethanol, loaded onto 5 ml Hitrap SP HP column (GE healthcare), and eluted with elution buffer (25 mM HEPES pH 7.5, 1 M NaCl, 10 mM 2-mercaptoethanol). Final purification was carried out using preparative grade HiLoad Superdex 200 26/600 (GE healthcare) with a buffer containing 25 mM HEPES pH 7.5, 250 mM NaCl, and 10 mM 2-mercaptoethanol. All Ubp10 constructs were purified using this protocol. Small aliquots were flash frozen using liquid nitrogen. Although the enzyme is very robust, we avoided freeze thawing for re-use.

#### Purification of wild-type histones

*Saccharomyces cerevisiae* histones H2A, H2B, H3, and H4 were expressed in *E.coli* and purified by standard methods (*Dyer et al., 2004*) with modifications as described previously (*Morgan et al., 2016*). All wild-type histone expression plasmids were generous gifts from the laboratory of Greg Bowman.

#### Preparation of non-hydrolyzable monoubiquitinated H2B

DUB-resistant monoubiquitinated yH2B containing a dichloroacetone linkage between ubiquitin and H2B-K123 (yH2B-DCA-Ub) was prepared using the approach previously described for *Xenopus* H2B-DCA-Ub (*Morgan et al., 2016*).

#### Purification of yeast FACT

All *Saccharomyces cerevisiae* FACT subunits, Nhp6 and the heterodimer of Spt16 and Pob3 were purified from yeast as previously described (*Paull and Johnson, 1995*; *Xin et al., 2009*).

### Preparation of cleavable monoubiquitinated histone H2B

Ubiquitinated yeast H2B was generated semi-synthetically according to protocols previously reported for *Xenopus* H2B-Ub (*Morgan et al., 2016*). In brief, ubiquitin (aa1-76, pMN43) and yH2B (aa1-119, pMN161) were cloned into pTXB1 (*Evans et al., 1998*; *Southworth et al., 1999*) by making C-terminal fusions with Mxe GyrA intein for thiol-induced cleavage and a chitin binding domain

(CBD) for affinity purification using chitin resin (NEB catalog #S6651L). By using this method, both the ubiquitin and H2B carrying a C-terminal reactive thioester are generated, which can then be used in subsequent ligation reactions. We made minor modifications to the previously used purification steps (Morgan et al., 2016). To purify ubiquitin, pMN43 (ubiquitin) was expressed in BL-21 RIL (DE3) cells. A 10-milliliter starter culture was transferred to 1 L 2xYT medium, incubated at 37°C until the OD reached 0.8, and induction was initiated with 0.5 mM IPTG at 16°C overnight. Pelleted cells were lysed in a buffer containing 100 mM NaOAc, 50 mM HEPES pH 6.5, 0.2 mM PMSF and 1 mM TCEP (Tris(2-carboxymethyl)phosphine hydrochloride). The lysate was spun down and the supernatant was loaded on to chitin resin, incubated for two hours to allow binding, and washed with lysis buffer. MES derivatization was initiated by adding cleavage buffer (100 mM NaOAc, 50 mM HEPES pH 6.5, 0.1 mM PMSF and 0.25 mM TCEP, and 250 mM MESNa (Sodium 2-Mercaptoethanesulfonate) and allowed to proceed at 37°C overnight. Cleavage was carried out over 5–6 rounds, each time collecting the derivatized ubiquitin by passing it through the chitin resin. The sample was then purified (twice) on an SP column (GE Healthcare), using binding buffer A (50 mM ammonium acetate pH 4.5, 0.5 mM TCEP) and eluted with 10% buffer B (50 mM ammonium acetate pH 4.5, 1 M NaCl, 0.5 mM TCEP). The final sample was thoroughly dialyzed against 0.5% TFA (Trifluoroacetic acid) and lyophilized. Derivatization was verified by MALDI-TOF mass spectrometry.

To purify histone H2B, pMN161 was expressed in BL-21 RIL (DE3) cells. A 10 ml starter culture was prepared from a freshly transformed plate, grown to an OD of 0.4, transferred and inoculated to an OD of 0.6–0.8 in 500 ml 2xYT medium, then induced overnight at 25°C by addition of 1 mM IPTG. Cells were pelleted by centrifugation, resuspended in 50 mM Tris-HCl pH 7.5, 200 mM NaCl, 1 mM EDTA, 0.1 mM PMSF, protease inhibitor tablet (1 tablet per 50 ml, Roche), and 0.25 mM TCEP, and lysed with a Microfluidizer (Microfluidics). To prevent degradation, the lysate was immediately spun down, the supernatant was applied on to pre-equilibrated chitin resin, and incubated at room temperature for 2–4 hr. The bound protein was washed with lysis buffer, followed by wash buffer 1 (50 mM Tris-HCl pH 7.2, 200 mM NaCl, 1 mM EDTA, 0.1 mM PMSF and 0.25 mM TCEP), and wash buffer 2 (50 mM Tris-HCl pH 7.4, 200 mM NaCl, 1 mM EDTA, 0.1 mM PMSF and 0.25 mM TCEP). MES derivatization was performed for 18 hr at 4°C in a buffer containing 50 mM Tris-HCl pH 7.4, 200 mM NaCl, 1 mM EDTA, 0.1 mM PMSF, 250 mM MESNa and 0.25 mM TCEP. Cleavage was terminated by adding 20 mM NaOAc pH 5.2, 7 M urea, 1 M NaCl, 1 mM EDTA, 0.1 mM PMSF and 0.25 mM TCEP. The protein was then purified by ion-exchange on an SP column (GE Healthcare) with binding buffer (20 mM NaOAc pH 5.2, 7 M urea, 1 mM EDTA, 0.1 mM PMSF and 0.25 mM TCEP) and eluted with a 0 to 1 M NaCl gradient. Following thorough dialysis against water, the protein was lyophilized and then resuspended in unfolding buffer (7 M Guanidinium-HCl, 20 mM Tris-HCl pH 7.5, and 10 mM DTT). The protein was then purified by HPLC on a C4 column (Higgins Analytical, PROTO C4 5 um 250 × 10 mm) equilibrated with 0.1% TFA and eluted with a 0% to 90% acetonitrile gradient elution. The final sample was then checked for derivatization by MALDI-TOF and immediately lyophilized to prevent hydrolysis.

Synthesis of the C-terminal H2B peptide and ligation of the peptide with H2B and Ub was performed as previously described (Morgan et al., 2016) with the following modifications. Briefly, synthetic peptide Cys-H2B(121-130) and purified thioester peptide H2B-(1-119)-MES were ligated using native chemical ligation, followed by unmasking the protected thiolysine with $MgCl_2$ and [Pd(Allyl) $Cl]_2$ (Jbara et al., 2016). The ligated product was then treated with DTT, purified via HPLC, and immediately lyophilized. Finally, H2B-(1-130) intermediate was ligated with Ub-MES prepared via intein method and the ligation product was subjected to a desulfurization step, which yielded the desired native H2B-Ub.

## Nucleosome reconstitution

### H2B-Ub containing nucleosomes

Histone octamers and a 146 bp DNA fragment containing the Widom 601 nucleosome positioning sequence were purified and reconstituted into nucleosomes using standard methods (Dyer et al., 2004). Nucleosomes containing H2B-DCA-Ub (non-hydrolyzable linkage) and H2B-Ub with the native isopeptide linkage were also reconstituted using the same method and purified using DEAE-5PW column (Tosoh Bioscience). Reconstituted nucleosomes were stored at 4°C and used as needed.

**Table 1.** Yeast Strains used All strains are congenic with the A364a background and are *MATa*.
Standard methods were used to introduce the mutations shown into diploid strains, then haploids were derived and crossed to obtain the combinations listed, ensuring that all strains with the same genotype displayed the phenotypes observed.

*Figure 4* Western blots

| Strain | Label | Genotype |
| --- | --- | --- |
| 8127-7-4 | WT | *ura3-Δ0 leu2-Δ0 trp1-Δtwo his3 lys2-128∂* |
| 10018-1-4 | *ubp10Δ* | *ura3-Δ0 leu2-Δ0 trp1-Δtwo his3 lys2-128∂ ubp10-Δ(::KanMX)* |
| 9204 | *pob3-L78R* | *ura3 leu2 trp1 his3 lys2-128∂ pob3-L78R(+34, LEU2)* |
| 10025-2-4 | *pob3-L78R ubp10Δ* | *ura3-Δ0 leu2-Δ0 trp1-Δtwo his3 lys2-128∂ ubp10-Δ(::HphMX) pob3-L78R(+34, LEU2)* |

*Figure 5A* (top panel), *Figure 6A* (1-4)

| | | |
| --- | --- | --- |
| 8127-7-4 | WT | *ura3-Δ0 leu2-Δ0 trp1-Δtwo his3 lys2-128∂* |
| 10018-1-4 | *ubp10Δ* | *ura3-Δ0 leu2-Δ0 trp1-Δtwo his3 lys2-128∂ ubp10-Δ(::KanMX)* |
| 9204 | *pob3-L78R* | *ura3 leu2 trp1 his3 lys2-128∂ pob3-L78R(+34, LEU2)* |
| 10025-2-4 | *pob3-L78R ubp10Δ* | *ura3-Δ0 leu2-Δ0 trp1-Δtwo his3 lys2-128∂ ubp10-Δ(::HphMX) pob3-L78R(+34, LEU2)* |

*Figure 5C* (middle panel), *Figure 6A* (5-8)

| | | |
| --- | --- | --- |
| 8127-7-4 | WT | *ura3-Δ0 leu2-Δ0 trp1-Δtwo his3 lys2-128∂* |
| 8540-1-1 | *ubp8Δ* | *ura3-Δ0 leu2-Δ0 trp1-Δtwo his3 lys2-128∂ ubp8-Δ(::KanMX)* |
| 9204 | *pob3-L78R* | *ura3 leu2 trp1 his3 lys2-128∂ pob3-L78R(+34, LEU2)* |
| 10032-4-3 | *pob3-L78R ubp8Δ* | *ura3-Δ0 leu2-Δ0 trp1-Δtwo his3 lys2-128∂ pob3-L78R(+34, LEU2) ubp8-Δ(::KanMX)* |

*Figure 5B* (bottom panel), *Figure 6B*

| | | |
| --- | --- | --- |
| 8127-7-4 | WT | *ura3-Δ0 leu2-Δ0 trp1-Δtwo his3 lys2-128∂* |
| 10018-1-4 | *ubp10Δ* | *ura3-Δ0 leu2-Δ0 trp1-Δtwo his3 lys2-128∂ ubp10-Δ(::KanMX)* |
| 9273H | *pob3-Q308K* | *ura3-Δ0 leu2-Δ0 trp1-Δtwo his3 lys2-128∂ pob3-Q308K(+34, HphMX)* |
| 10019-2-3 | *pob3-Q308K ubp10Δ* | *ura3-Δ0 leu2-Δ0 trp1-Δtwo his3 lys2-128∂ pob3-Q308K(+34, HphMX) ubp10-Δ(::KanMX)* |
| 9495 H-2-3 | *spt16-11* | *ura3 leu2 trp1 his3 lys2-128∂ spt16-11(+124, HphMX)* |

*Figure 6C*

| | | |
| --- | --- | --- |
| 9880-2-2 | WT | *ura3-Δ0 leu2-Δ0 trp1-Δtwo his3 lys2-128∂ GAL1pr-flo8-HIS3(NatMX)* |
| 10040-3-2 | *ubp8Δ* | *ura3-Δ0 leu2-Δ0 trp1-Δtwo his3 lys2-128∂ GAL1pr-flo8-HIS3(NatMX) ubp8-Δ(::KanMX)* |
| 10024-3-1 | *ubp10Δ* | *ura3 leu2-Δ0 trp1 his3 lys2-128∂ GAL1pr-flo8-HIS3(NatMX) ubp10-Δ(::HphMX)* |
| 10040-1-3 | *pob3-L78R* | *ura3-Δ0 leu2-Δ0 trp1-Δtwo his3 lys2-128∂ GAL1pr-flo8-HIS3(NatMX) pob3-L78R(+34, LEU2)* |
| 10040-5-1 | *pob3-L78R ubp8Δ* | *ura3-Δ0 leu2-Δ0 trp1-Δtwo his3 lys2-128∂ GAL1pr-flo8-HIS3(NatMX) pob3-L78R(+34, LEU2) ubp8-Δ(::KanMX)* |
| 10039-1-4 | *pob3-L78R ubp10Δ* | *ura3-Δ0 leu2-Δ0 trp1-Δtwo his3 lys2-128∂ GAL1pr-flo8-HIS3(NatMX) pob3-L78R(+34, LEU2) ubp10-Δ(::HphMX)* |
| 9949-3-1 | *spt16-11* | *ura3 leu2 trp1 his3 lys2-128∂ GAL1pr-flo8-HIS3(NatMX) spt16-11(+124, KanMX)* |
| 10044-4-2 | *ubp8Δ spt16-11* | *ura3-Δ0 leu2-Δ0 trp1-Δtwo his3 lys2-128∂ GAL1pr-flo8-HIS3(NatMX) spt16-11 ubp8-Δ(::KanMX)* |

*Table 1 continued on next page*

*Table 1 continued*

*Figure 4 Western blots*

| Strain | Label | Genotype |
|---|---|---|
| 10043-7-3 | *ubp10∆ spt16-11* | *ura3-∆0 leu2-∆0 trp1-∆two his3 lys2-128∂ GAL1pr-flo8-HIS3(NatMX) spt16-11(+124, HphMX) ubp10-∆(::KanMX)* |

**11-16-2018 upb10∆(109-133) tests (Figure 6—figure supplement 1)**

| | | |
|---|---|---|
| 8127-7-4 | WT | *ura3-∆0 leu2-∆0 trp1-∆two his3 lys2-128∂* |
| 10018-1-4 | *ubp10∆* | *ura3-∆0 leu2-∆0 trp1-∆two his3 lys2-128∂ ubp10-∆(::KanMX)* |
| 9204 | *pob3-L78R* | *ura3 leu2 trp1 his3 lys2-128∂ pob3-L78R(+34, LEU2)* |
| 10062-4-4 | *pob3-L78R ubp10∆* | *ura3-∆0 leu2-∆0 trp1-∆two his3 lys2-128∂ ubp10-∆(::HphMX) pob3-L78R(+34, LEU2)* |
| 10064-2-1 | *ubp10∆(109-133)* | *ura3-∆0 leu2-∆0 trp1-∆two his3 lys2-128∂ ubp10-∆(109-133)* |
| 10062-1-2 | *pob3-L78R ubp10∆(109-133)* | *ura3-∆0 leu2-∆0 trp1-∆two his3 lys2-128∂ pob3-L78R(+34, LEU2) ubp10-∆(109-133)* |

DOI: https://doi.org/10.7554/eLife.40988.018

## Flag-tagged yeast nucleosomes

Flag-tagged histone octamers were purified from *E.Coli* using a polycistronic expression vector containing all four yeast histones (a generous gift from Alwin Köhler) and a purification tag on H2B as previously described (*Turco et al., 2015*). Nucleosomes were reconstituted and purified using standard methods.

## Electrophoretic mobility shift assays (EMSAs)

Ubiquitinated or wild-type nucleosomes (100 nM) and Ubp10 concentrations ranging from 0 to 1600 nM were incubated on ice for 1 hr in the presence of 20 mM HEPES pH 7.6, 50 mM NaCl, 5% sucrose, 1 mM DTT, and 2.5 mM $MgCl_2$ and 0.1 mg/ml bovine serum albumin (BSA). Once the reaction was completed, the samples were immediately loaded on to a pre-run 6% Novex TBE gels (Life Technologies) and electrophoresed for 75–100 min using 0.25x TBE running buffer at 4°C. Gels were stained with SYBR gold (Life Technologies) for 20 min and imaged using Chemidoc Touch (Bio-Rad). Apparent dissociation constants were estimated from half-maximal Ubp10-nucleosome complexes on native gel.

## Ubp10 deubiquitination activity assays

Deubiquitination activity assays were performed according to a previously described protocol (*Morgan et al., 2016*). Briefly, 1 μM yNCP-Ub and 2 μM yH2A/H2B-Ub were preincubated in a 30°C water bath for 30 min in DUB assay buffer (50 mM HEPES pH 7.6, 150 mM NaCl, and 5 mM DTT). Isopeptidase activity was initiated by adding 5 nM pre-warmed (5 min) Ubp10. Similar concentrations were used for the experiments involving Ubp10 truncations. For experiments involving FACT, 2 μM Nhp6 and 2 μM Spt16/Pob3 were pre-incubated with the substrates. Time-courses were monitored by removing samples at the indicated times and quenching the reactions with 1x-LDS (Bio-Rad). Samples were analyzed on commercial SDS-PAGE gels (NuPAGE and Criterion) stained with SYPRO Ruby and imaged with Chemidoc Touch (Bio-Rad). All experiments were carried out in siliconized low retention tubes (Fisher Scientific Cat. No.02-681-320).

## Yeast growth assays

Yeast strains with the genotypes shown in *Table 1* were grown to saturation in rich medium then 10-fold serial dilutions were spotted to agarose plates with the composition described in *Figure 5*.

## Western blots

Western blots were performed as in *McCullough et al. (2018)* using the TCA method of protein extraction. Each gel contained a dilution series of the WT strain extract to establish linearity of response and to determine the concentration of the target protein.

## Acknowledgements

We thank members of the Wolberger lab for advice and feedback. We are grateful to Anthony DiBello for initially cloning Ubp10 into a bacterial expression plasmid. Funding was provided by National Institutes of Health grants GM095822 (CW) and GM064649 (TF), the Jordan and Irene Tark Academic Chair (AB), an Israel Council of Higher Education Fellowship (MJ), and a National Science Foundation Graduate fellowship (MN).

## Additional information

### Competing interests

Cynthia Wolberger: Reviewing editor, *eLife*. Tim Formosa: Reviewing editor, *eLife*. The other authors declare that no competing interests exist.

### Funding

| Funder | Grant reference number | Author |
| --- | --- | --- |
| National Institute of General Medical Sciences | GM095822 | Cynthia Wolberger |
| National Institute of General Medical Sciences | GM064649 | Tim Formosa |
| National Science Foundation | Graduate Research Fellowship | Melesse Nune |
| Jordan and Irene Tark Academic Chair | | Ashraf Brik |
| Israel Council of Higher Education | Fellowship | Muhammad Jbara |
| National Institute of General Medical Sciences | Training Grant GM008403 | Melesse Nune |

The funders had no role in study design, data collection and interpretation, or the decision to submit the work for publication.

### Author contributions

Melesse Nune, Michael T Morgan, Tim Formosa, Cynthia Wolberger, Conceptualization, Formal analysis, Supervision, Funding acquisition, Investigation, Methodology, Writing—original draft, Project administration, Writing—review and editing; Zaily Connell, Muhammad Jbara, Resources, Investigation, Methodology; Laura McCullough, Supervision, Investigation, Methodology; Hao Sun, Investigation, Involved in the synthesis of mercaptolysine amino acid, a critical reagent for semisynthesis of H2Bub; Ashraf Brik, Conceptualization, Supervision, Investigation, Methodology, Writing—review and editing

### Author ORCIDs

Melesse Nune (iD) http://orcid.org/0000-0002-3385-3320
Tim Formosa (iD) http://orcid.org/0000-0002-8477-2483
Cynthia Wolberger (iD) http://orcid.org/0000-0001-8578-2969

### Decision letter and Author response

Decision letter https://doi.org/10.7554/eLife.40988.021
Author response https://doi.org/10.7554/eLife.40988.022

## Additional files

### Supplementary files
• Transparent reporting form
DOI: https://doi.org/10.7554/eLife.40988.019

### Data availability
All data generated or analyzed during this study are included in the manuscript and supporting files.

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
