## [Decision Letter]

Thank you for submitting your article "FACT and Ubp10 collaborate to modulate H2B deubiquitination and nucleosome dynamics" for consideration by *eLife*. Your article has been reviewed by two peer reviewers, and the evaluation has been overseen by a Geeta Narlikar as Reviewing Editor and Jessica Tyler as the Senior Editor. The following individuals involved in review of your submission have agreed to reveal their identity: Vasily M Studitsky (Reviewer #2).

The reviewers have discussed the reviews with one another and the Reviewing Editor has drafted this decision to help you prepare a revised submission. As you will see the reviewers agree that your work contains new experimental findings that are important for understanding the role of H2B-Ub and histone chaperone FACT in chromatin dynamics during transcription and replication. However they also have some comments that they would like you to address experimentally to strengthen your core conclusions.

Summary:

Histone H2B ubiquitination is a highly conserved PTM coupled to active transcription. Although the enzymes that are involved in the ubiquitination and deubiquitination processes are largely known through genetics, the highly dynamic nature of this PTM has made mechanistic understanding of its function very difficult.

This manuscript reports the surprising finding that histone chaperone FACT is a key regulator of H2B deubiquitinase Ubp10 in yeast. In vitro, FACT strongly stimulates Ubp10 to deubiquitinate nucleosomes, although the underlying mechanism is not entirely clear. In vivo, *ubp10*-Δ and FACT mutants show synthetic effects in HU sensitivity and activation of a cryptic transcription reporter gene, suggesting that they function together in replication and transcription. Since Ubp10 was previously thought to function at subtelomere regions, these results revealed a possible new function of Ubp10 in transcriptionally active coding regions and provide a missing piece of H2Bub dynamics.

Overall the data are of high quality and presented with transparency. The results will be of interest to a broad audience in the transcription/chromatin field. However some key experimental tests are required to strengthen the conclusions as described below. Some textual clarifications are also required.

Essential revisions:

1) In Figure 2B: 2 μM Spt16/Pob3-WT and 2 μM Nhp6 was used. Is the nucleosome reorganized under these conditions? According to past studies from Formosa lab, typically reorganization requires at least 6-fold molar excess of Nhp6. More importantly does the stimulatory effect of FACT on Ubp10 action require both Spt16/Pob3 and Nhp6 in vitro? Or does FACT facilitate the deubiquitination in the absence of Nhp6 (and thus in the absence of nucleosome reorganization)? This is a critical experiment to further validate the model proposed in Figure 7.

2) The very detailed proposed model suggests that nucleosome reorganization is necessary for the deubiquitination; however, this remains to be rigorously established (see comment 1). Furthermore, nucleosome reorganization requires the presence of Nhp6 and accordingly the model explicitly implicates Nhp6 in nucleosome reorganization. However, the cycles of ubiquitination/deubiquitination are presumably coupled to ongoing transcription/replication and Nhp6 is generally not present on transcribed genes. Therefore overall, the model should be further clarified and better connected to transcription and replication.

3) Because Ubp10 has known to function in silencing, there is a formal possibility that *ubp10*-Δ leads to nonspecific spread of the SIR complex, which causes the observed genetic defects. Since the N-terminal regions of Ubp10 is required for its function at subtelomeres, and appears to be dispensable for FACT-dependent activity in vitro, the authors should rescue the synthetic genetic effects with Ubp10 N-terminal truncation mutants.

4) The result in Figure 4 is surprising. Previously it was reported that inactivating FACT (using a different allele) eliminates H2Bub (Fleming et al., 2008). It's possible that this difference is due to allele-specific effects. However, higher levels H2Bub have been shown to associate with better nucleosome assembly (Batta et al., 2011). Figure 6 instead suggests that nucleosome assembly is defective. This contradiction needs to be addressed with more direct evidence (see point 5 below).

5) Since westerns are notoriously difficult in discerning small differences, a more direct approach would be ChIP analysis of H2B and H2Bub levels across, for example, a long gene previously shown affected by Ubp10. This type of analysis would provide more mechanistic insights as to the effect on nucleosome integrity and relative H2Bub levels (normalized to H2B). It will also separate the effects of ubp10delta at subtelomere vs actively transcribed regions.

6) The authors propose that FACT can stimulate Ubp10 action on H2B ubiquitylated nucleosomes by either partially unfolding the nucleosome or by dissociating the H2A/H2B dimer. A straightforward way to test for either possibility would be to run the deubiquitylated products on a native gel. If the products run analogous to unmodified nucleosomes this will suggest that FACT transiently unravels the nucleosome but that the nucleosome refolds after deubiquitylation. Instead if the products run analogous to tetrasomes or hexasomes, this will suggest that Ubp10 acts on a free H2A/H2B dimer that is released by FACT action.

7) While the synthetic interaction is consistent with such a possibility, there is no direct evidence that FACT and Ubp10 function together in vivo. So we suggest that the authors make a statement in the Discussion such as: "While there is no direct evidence that FACT and Ubp10 work together in vivo, the synthetic genetic interaction and the biochemical cooperation that we observe is consistent with such a possibility".

---

## [Author Response]

Essential revisions:1) In Figure 2B: 2 μM Spt16/Pob3-WT and 2 μM Nhp6 was used. Is the nucleosome reorganized under these conditions? According to past studies from Formosa lab, typically reorganization requires at least 6-fold molar excess of Nhp6. More importantly does the stimulatory effect of FACT on Ubp10 action require both Spt16/Pob3 and Nhp6 in vitro? Or does FACT facilitate the deubiquitination in the absence of Nhp6 (and thus in the absence of nucleosome reorganization)? This is a critical experiment to further validate the model proposed in Figure 7.

We thank the reviewers for suggesting this experiment. Since yeast Spt16-Pob3 does not bind stably to nucleosomes and is unable to induce nuclease sensitivity without added Nhp6, we had assumed its effects on Ubp10 activity would also have this feature. As shown in the new Figure 2—figure supplement 2 and 4, this assumption was incorrect, as Spt16-Pob3 was able to stimulate Ubp10 activity on nucleosomal H2B-Ub nearly as efficiently without Nhp6 as with it. This has implications for the model regarding how FACT affects Ubp10 activity, and we have made corresponding changes to Figure 7 as a result.

Perhaps more importantly, this result provides more insight into how FACT affects nucleosomal structure, increasing the impact of the results. Human FACT and yeast Spt16-Pob3 do not stably alter nucleosome structure, but each is able to do so with added Nhp6. However, Nhp6 is not essential for viability in yeast, whereas Spt16 and Pob3 are, and human FACT affects RNA Pol II passage under conditions where it does not produce full reorganization. It is therefore clear that Nhp6 is not required for FACT activity, but supports it in vivo and in vitro. Several labs have shown that FACT has multiple domains that interact with histone surfaces, supporting a model in which FACT binds to nucleosomes in several modes. It seems unlikely that producing the fully reorganized form detected by EMSA or restriction endonuclease digestion is the only relevant function of FACT, and we previously proposed a transient "assembly checkpoint" form of FACT:nucleosome complexes to explain some genetic and biochemical results (McCullough, et al., 2018). We now propose that one of these modes of binding along the pathway to (or returning from) full reorganization is the form that is relevant for enhancing Ubp10 activity, thus explaining the lack of a requirement for Nhp6 to produce the stably reorganized form. This is significant because it reveals a function for a partially disrupted form of nucleosomes, and because it diminishes the possibility that the stimulation of Ubp10 activity is a simple consequence of H2A-H2B dimer eviction, which does not occur in the absence of Nhp6.

2) The very detailed proposed model suggests that nucleosome reorganization is necessary for the deubiquitination; however, this remains to be rigorously established (see comment 1). Furthermore, nucleosome reorganization requires the presence of Nhp6 and accordingly the model explicitly implicates Nhp6 in nucleosome reorganization. However, the cycles of ubiquitination/deubiquitination are presumably coupled to ongoing transcription/replication and Nhp6 is generally not present on transcribed genes. Therefore overall, the model should be further clarified and better connected to transcription and replication.

The lack of an absolute requirement for Nhp6 in the stimulation of Ubp10 activity on nucleosomal substrates changes our model and our discussion of it as discussed above under point 1. We have also modified the model figure accordingly.

3) Because Ubp10 has known to function in silencing, there is a formal possibility that ubp10-Δ leads to nonspecific spread of the SIR complex, which causes the observed genetic defects. Since the N-terminal regions of Ubp10 is required for its function at subtelomeres, and appears to be dispensable for FACT-dependent activity in vitro, the authors should rescue the synthetic genetic effects with Ubp10 N-terminal truncation mutants.

Richard Gardner's lab identified Ubp10 residues 109-133 as the region necessary for interaction with the SIR complex and showed that the *ubp10-∆(109-133)* allele exhibited a silencing defect comparable to *ubp10∆* (Reed et al., 2015). We deleted these residues from the native *UBP10* locus using a markerless conversion method (Storici et al., Nat Biotechnol, 2001 19:773), combined the *ubp10-∆(109-133)* allele with FACT mutations, and repeated our genetic analysis. The mutation lacking the SIR-interaction domain did not affect the properties of FACT mutants in the manner observed for the full deletion of *UBP10*, ruling out the possibility that the phenotypes we observe are caused by the function of Ubp10 in silencing. We show the results for the *pob3-L78R* allele in Figure 6—figure supplement 1.

4) The result in Figure 4 is surprising. Previously it was reported that inactivating FACT (using a different allele) eliminates H2Bub (Fleming et al., 2008). It's possible that this difference is due to allele-specific effects. However, higher levels H2Bub have been shown to associate with better nucleosome assembly (Batta et al., 2011). Figure 6 instead suggests that nucleosome assembly is defective. This contradiction needs to be addressed with more direct evidence (see point 5 below).

We have found that strains with the *spt16-G132D* allele used by Fleming et al. have normal levels of Pob3 but reduced Spt16 under permissive conditions, whereas *pob3-L78R* strains have about a 10-fold reduction in Pob3 but nearly normal levels of Spt16 (VanDemark, et al., 2008). Both alleles therefore examine cells with reduced FACT levels; however, one goes from normal to very low FACT (*spt16-G132D = spt16-197; acute withdrawal*) while the other has chronically low FACT throughout the experiment. Further, the balance of Spt16 to Pob3 is expected to be different in the two strains. Since complete withdrawal of FACT is ultimately lethal, we think the use of chronic reduction under viable conditions is preferable to complete inactivation of Spt16 at a non-permissive temperature. Overall, we do not know why Fleming et al. observed decreased FLAG-H2B-ub levels under conditions of lethal withdrawal of FACT using the *spt16-G132D* allele, but we are confident that H2B-Ub increases under chronic Pob3 depletion with *pob3-L78R*. Moreover, we claim that *pob3L78R* affects Ubp10 activity in vivo, not that all FACT defects do. We have added these ideas to the manuscript (subsection “A FACT mutant strain has elevated levels of H2B-Ub”, last paragraph).

The data from Batta et al., 2011, show a more complex picture than a simple association of higher H2B-Ub with greater nucleosome occupancy. While that study shows a global decrease in nucleosome occupancy for an H2B-K123A mutation, that mutant cannot undergo cycles of H2B ubiquitination/deubiquitination and is presumably defective in FACT recruitment, all of which are thought to be important for maintaining chromatin organization (Batta et al., 2011; Fleming et al., 2008; Henry et al., 2003; Pavri et al., 2006). The decrease in global nucleosome occupancy observed in deletions of the H2B E2 ubiquitin conjugating enzyme (Rad6) and associated factor (Lge1, which binds to the E3 ligase, Bre1), and increase in global nucleosome occupancy observed in a *ubp8*D deletion is consistent with a role for H2B-Ub in stabilizing nucleosomes (the effects of *ubp10*D were not examined). However, the simple model that high global levels of H2BUb correlate with higher nucleosome occupancy is contradicted by their observation that highly transcribed genes have normal levels of nucleosome occupancy in a *ubp8∆* mutant, even though these genes have markedly lower nucleosome occupancy in *rad6∆* and H2B-K123A mutants (Batta et al., 2011). The relationship between global H2B-Ub levels and nucleosome occupancy is thus clearly complex. Our model that both ubiquitination and deubiquitination are important for maintaining chromatin structure and that Ubp10 and FACT function together in this process is consistent the general consensus that cycles of H2B ubiquitination and deubiquitination are critical to maintaining chromatin organization. We have added a paragraph to the Discussion (third paragraph) that summarizes these points.

5) Since westerns are notoriously difficult in discerning small differences, a more direct approach would be ChIP analysis of H2B and H2Bub levels across, for example, a long gene previously shown affected by Ubp10. This type of analysis would provide more mechanistic insights as to the effect on nucleosome integrity and relative H2Bub levels (normalized to H2B). It will also separate the effects of ubp10delta at subtelomere vs actively transcribed regions.

We believe that the western blot data we are presenting meet a much higher standard of stringency than those presented in the Fleming paper (see response to point 4 above). We agree that quantitation of western blots with histones can be difficult, but we note that we used stringent methods including a dilution series on every blot to establish linearity of all responses (with an example shown in our figure), and we report quantitative results from at least 13 independent replicates. This is in contrast to the result reported by Fleming et al. who based their conclusion on visual inspection of a single lane from a single western without quantitation or normalization. Further, their strain carried a deletion of the two H2B genes covered with a plasmid expressing FLAG-tagged H2B (H2B-FLAG). It also had a deletion of *SPT16* covered with an *spt16-G132D* plasmid; our published work shows that each of these manipulations can lead to misregulation of at least some genes, so we instead used modifications of native loci in our experiments (unperturbed H2B genes with H2B and H2B-Ub detected by separate specific antibodies, and an otherwise unperturbed *POB3* locus with just the L78R mutation.) We are therefore much more confident that this allele of FACT leads to elevated H2B-Ub than we are confident that *spt16-G132D* grown at the non-permissive temperature leads to its loss.

We think that western blots provide the appropriate global information we need to ask a question about the effects of Ubp10, which we propose acts genome-wide for producing global chromatin architecture, whereas Ubp8 acts locally in the context of transcription. Western blots measure the global level of H2B-Ub whereas ChIP measures local levels, which are known to vary at different sites relative to the transcription start site, as discussed in several places in this manuscript. Using ChIP to answer this question assumes that we know which loci should be affected and which should not, but this in turn assumes a model in which ubiquitination exclusively supports nucleosome stability and that loss of Ubp10 will have predictable effects localized to the same genes whose transcription is altered by this mutation. As discussed below, these assumptions may not be valid, and testing them is beyond the scope of this manuscript. Our claim is that a FACT defect can cause global loss of deubiquitination of H2B by Ubp10; we do not know or claim to know where this failure occurs.

To address the reviewers’ suggestion, we used anti-H2B-Ub ChIP to test several genes (PMA1, the HO promoter, 5' end, and 3' end, and PHO89) of different sizes whose transcript levels change in different ways in a *ubp10∆* strain, but we did not see a uniform pattern of effects. H2B-Ub levels were unaffected by any mutation in PMA1, they were elevated in the promoter and 5' end of the *HO* gene in a *pob3-L78R* strain but not in a *ubp10∆* strain, and they increased at the 3' end of *HO* in both mutants. Selected regions therefore support the model that FACT collaborates with Ubp10, but the story is clearly more complex. The initial rationale for attempting this approach was that it is difficult to use western blots to detect small changes in H2B-Ub, but ChIP introduces variation from both the antibody capture of material and using exponential PCR to detect small changes. We used 4 biological replicates of each strain and 4 technical replicates of each IP in our tests and observed more variation among the results than we did with western blots, not less. We therefore conclude that our western blot data meet the highest standard of stringency for this method, answer the question we are asking, and are more reliable for this purpose than ChIP.

The ChIP-qPCR approach is also less suitable for our needs than westerns because there is no suitable commercially available anti-H2B antibody available for ChIP, so we cannot normalize the H2B-Ub levels to the H2B levels without resorting to using an epitope tag that disrupts the phenomenon we are measuring. Our tests show that supplying FLAG-tagged H2B affects the phenotypes of FACT mutants, making this an unacceptable alternative. Overall, the only way to pursue the interesting effects we have observed is to do a ChIP-seq experiment and compare the outcomes to nucleosome occupancy and positioning by MNase-seq. We are considering initiating these studies, but they are clearly beyond the scope of this revision.

6) The authors propose that FACT can stimulate Ubp10 action on H2B ubiquitylated nucleosomes by either partially unfolding the nucleosome or by dissociating the H2A/H2B dimer. A straightforward way to test for either possibility would be to run the deubiquitylated products on a native gel. If the products run analogous to unmodified nucleosomes this will suggest that FACT transiently unravels the nucleosome but that the nucleosome refolds after deubiquitylation. Instead if the products run analogous to tetrasomes or hexasomes, this will suggest that Ubp10 acts on a free H2A/H2B dimer that is released by FACT action.

We have performed the suggested experiment, which is shown in Figure 2—figure supplement 4. The results show that the presence of FACT during the deubiquitination assay has no effect on the mobility of the three nucleosome species (with 2, 1 or 0 ubiquitins), indicating that the nucleosomes remain intact. This result suggests that FACT's effects on the nucleosomes that stimulate the activity of Ubp10 are transient and reversible.

7) While the synthetic interaction is consistent with such a possibility, there is no direct evidence that FACT and Ubp10 function together in vivo. So we suggest that the authors make a statement in the Discussion such as: "While there is no direct evidence that FACT and Ubp10 work together in vivo, the synthetic genetic interaction and the biochemical cooperation that we observe is consistent with such a possibility".

This statement has been added to the Discussion.